# Learning Provably Robust Estimators for Inverse Problems via Jittering

**Anselm Krainovic**
Technical University of Munich
`anselm.krainovic@tum.de`

**Mahdi Soltanolkotabi**
University of Southern California
`soltanol@usc.edu`

**Reinhard Heckel**
Technical University of Munich
`reinhard.heckel@tum.de`

## Abstract

Deep neural networks provide excellent performance for inverse problems such as denoising. However, neural networks can be sensitive to adversarial or worst-case perturbations. This raises the question of whether such networks can be trained efficiently to be worst-case robust. In this paper, we investigate whether jittering, a simple regularization technique that adds isotropic Gaussian noise during training, is effective for learning worst-case robust estimators for inverse problems. While well studied for prediction in classification tasks, the effectiveness of jittering for inverse problems has not been systematically investigated. In this paper, we present a novel analytical characterization of the optimal $\ell_2$-worst-case robust estimator for linear denoising and show that jittering yields optimal robust denoisers. Furthermore, we examine jittering empirically via training deep neural networks (U-nets) for natural image denoising, deconvolution, and accelerated magnetic resonance imaging (MRI). The results show that jittering significantly enhances the worst-case robustness, but can be suboptimal for inverse problems beyond denoising. Moreover, our results imply that training on real data which often contains slight noise is somewhat robustness enhancing.

## 1 Introduction

Deep neural networks achieve state-of-the-art performance for image reconstruction tasks including compressive sensing, super-resolution, and denoising. Due to their excellent performance, deep networks are now used in a variety of imaging technologies, for example in MRI and CT. However, concerns have been voiced that neural networks can be sensitive to worst-case or adversarial perturbations. Those concerns are fuelled by neural networks being sensitive to small, adversarially selected perturbations for prediction tasks such as image classification (Szegedy et al., 2014).

Recent empirical work for image reconstruction tasks (Huang et al., 2018; Antun et al., 2020; Genzel et al., 2022; Darestani et al., 2021) found that worst-case perturbations can have a significantly larger effect on the image quality than random perturbations. This sensitivity to worst-case perturbations is not unique to neural networks, classical imaging methods are similarly sensitive (Darestani et al., 2021).

This raises the question of whether networks can be designed or trained to be worst-case robust. A successful method proposed in the context of classification is adversarial training, which optimizes a robust or adversarial loss during training (Madry et al., 2018). However, the robust loss requires finding worst-case perturbations during training which is difficult and computationally expensive.

37th Conference on Neural Information Processing Systems (NeurIPS 2023).

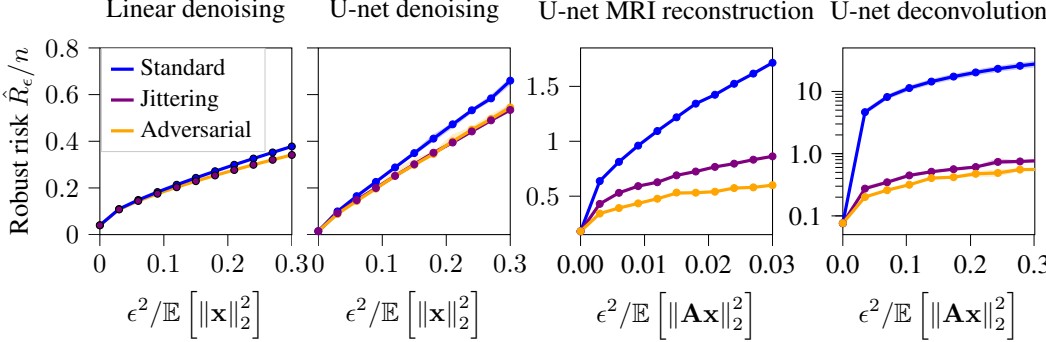

Figure 1: **Jittering yields worst-case robust reconstruction methods.** The plots show the pixel-wise empirical robust risks $\hat{R}_\epsilon/n$ of models trained to minimize the robust risk $R_\epsilon$ and jittering risk $J_{\sigma_w}$, respectively, with suitable choices of jittering levels $\sigma_w(\epsilon)$. The optimal robust risk is approximated via adversarial training. The shaded areas are $66\%$ confidence intervals. Left panel is for subspace denoising where jittering and robust training are provably equivalent, the panels from second left to right are for image reconstruction problems with the U-net, where jittering is particularly effective for denoising.

In this work, we study jittering, a simple regularization technique that adds noise during training as a robustness-enhancing technique for inverse problems. It is long known that adding noise during training has regularizing effect and can be beneficial for generalization (Bishop, 1995; Holmstrom and Koistinen, 1992; Reed and Marks II, 1999). Prior work also studied adding noise for enhancing adversarial robustness for classification (Zantedeschi et al., 2017; Kannan et al., 2018; Gilmer et al., 2019). However, jittering has not been systematically studied as a robustness enhancing technique for training robust networks for inverse problems.

We consider the following signal reconstruction problem. Let $f\colon \mathbb{R}^m \to \mathbb{R}^n$ be an estimator (a neural network in practice) for a signal (often an image) $\mathbf{x} \in \mathbb{R}^n$ based on the measurement $\mathbf{y} = \mathbf{A}\mathbf{x} + \mathbf{z} \in \mathbb{R}^m$, where $\mathbf{A}$ is a measurement matrix and $\mathbf{z}$ is random noise. We want to learn an estimator that has small robust risk defined as

$$R_\epsilon(f) = \mathbb{E}_{(\mathbf{x},\mathbf{y})} \left[ \max_{\|\mathbf{e}\|_2 \leq \epsilon} \|f(\mathbf{y} + \mathbf{e}) - \mathbf{x}\|_2^2 \right]. \tag{1}$$

The robust risk is the expected worst-case error with respect to a $\ell_2$-perturbation of norm at most $\epsilon$ of $f$ measured with the mean-squared error.

**Theoretical results.** We start with Gaussian denoising of a signal lying in a subspace, and first characterize the optimal linear robust denoiser, i.e., the estimator that minimizes the robust risk $R_\epsilon(f)$. While the resulting estimator is quite intuitive, proving optimality is fairly involved and relies on interesting applications of Jensen's inequality.

Second, we show that the optimal linear robust estimator minimizes the Jittering-risk

$$J_{\sigma_w}(f) = \mathbb{E}_{(\mathbf{x},\mathbf{y}),\mathbf{w}} \left[ \|f(\mathbf{y} + \mathbf{w}) - \mathbf{x}\|_2^2 \right], \tag{2}$$

where $\mathbf{w} \sim \mathcal{N}(0, \sigma_w^2 \mathbf{I})$ is Gaussian jittering noise with noise variance $\sigma_w^2$ that depends on the desired robustness level $\epsilon$. This finding implies that instead of performing robust training via minimizing an empirical version of robust risk, we can train a denoiser via jittering, i.e., injecting Gaussian noise during training, at least for denoising a signal lying in a subspace. Figure 1, left panel, demonstrates the equivalence of training via minimizing a jittering risk and robust training numerically for the subspace model. It is evident that both methods of training yield an equally robust estimator.

Moreover, we discuss extensions of our theory for linear inverse problems $\mathbf{y} = \mathbf{A}\mathbf{x} + \mathbf{z}$ and find that jittering can result in suboptimal worst-case robust estimators for some classes of forward operators.

**Empirical results for real-world denoising, deconvolution and compressive sensing.** Jittering is also effective for learning robust neural network estimators for solving inverse problems in practice. Figure 1, second from left to right, depicts the worst-case risk achieved by training a U-net model for denoising, compressive sensing, and deconvolution, with standard training (blue), with jittering (purple), and with adversarial training (orange). For denoising, we see that jittering is as effective for obtaining a worst-case robust estimator as adversarial training, as suggested by theory. For compressive sensing and deconvolution, we find that jittering can be suboptimal beyond denoising, but is still effective for enhancing robustness.

Those findings make jittering a potentially attractive method for learning robust estimators in the context of inverse problems, since jittering can also be implemented easily and needs far less computational resources than adversarial training. Moreover, those findings imply that training on real data which often contains slight noise is somewhat robustness enhancing.

## 2 Related work

**Empirical investigation of worst-case robustness for imaging.** Several works investigated the sensitivity of neural networks for image reconstruction tasks to adversarial perturbations, for limited angle tomography (Huang et al., 2018), MRI and CT (Antun et al., 2020; Genzel et al., 2022; Darestani et al., 2021), and image-to-image tasks (colorization, deblurring, denoising, and super-resolution) (Choi et al., 2022; Yan et al., 2022; Choi et al., 2019). Collectively, those works show that neural networks for imaging problems are significantly more sensitive to adversarial perturbations than to random perturbations, as expected. The effect of adversarial $\ell_2$-perturbations measured in mean-squared-error is roughly proportional to the energy of the perturbations in most of those problems, demonstrating that up to a constant (that might be large) neural networks can be relatively stable for imaging tasks. Classical reconstruction methods, in particular $\ell_1$-regularized least-squares, are similarly sensitive to adversarial perturbations (Darestani et al., 2021).

**Learning robust methods with robust optimization.** To learn robust classifiers, Madry et al. (2018) proposed to minimize a robust loss and to find worst-case perturbations during training with projected gradient descent. Adversarial training can be effective for learning robust methods, but is computationally expensive due to the cost of finding adversarial perturbations. A variety of heuristics exist to lower the computational cost of robust training for neural networks. For example, Raj et al. (2020) consider compressive sensing and CT reconstruction problems and propose to generate adversarial perturbations for training with an auxiliary network instead of solving a maximization problem. As another example, Wong et al. (2020) considers adversarial training of classifiers and propose to calculate adversarial perturbations during training by first randomly perturbing the initial point and then applying a single step of projected gradient descent.

**Jittering for enhancing robustness in inverse problems.** The literature is somewhat split on whether jittering is effective for enhancing worst-case robustness for imaging. Genzel et al. (2022) suggested that jittering can enhance worst-case robustness of networks trained on compressive sensing and CT tasks. Contrary, Gandikota et al. (2022) consider the robustness to $\ell_\infty$-perturbations for neural-network based deblurring and observed that the DeepWiener architecture, trained via Jittering with noise levels sampled from a fixed interval, is sensitive to adversarial perturbations. In this work, we present the first systematic study on the effectiveness of jittering as a robustness-enhancing technique for inverse problems.

**Robustness for inverse problems versus robustness for classification problems.** Robustness in general and adding noise during training in particular, has been intensively studied in the classification setting. However, inverse problems and classification/prediction problems are very different. Adversarial robustness for classifiers is defined as the (average) minimal distance to the decision boundary, and random noise robustness as the minimal noise strength (for example the radius of Gaussian noise sphere) such that one likely crosses the decision boundary (Fawzi et al., 2018). For inverse problems, there is no notion of a decision boundary. Therefore, results and intuitions from classification, which are often based on geometric insights on distances to surfaces (see for example Fawzi et al. (2016) and Shafahi et al. (2019)) do not apply to inverse problems.

**Jittering for enhancing robustness in classification.** Prior work in classification considered Gaussian data augmentation or adding Gaussian noise during training (which is conceptually very similar to jittering) as an robustness-enhancing technique and found that adding noise enhances adversarial robustness, but reported mixed results on its effectiveness. Fawzi et al. (2018) proved for linear classifiers that adding Gaussian noise during training increases adversarial robustness, and Gilmer et al. (2019) demonstrated that empirically adding Gaussian noise during training also increases adversarial robustness for neural networks in the context of classification. Rusak et al. (2020) also found Gaussian noise addition beneficial for corruption robustness (including noise, compression and weather artifacts). Furthermore, Kannan et al. (2018) and Zantedeschi et al. (2017) considered adding Gaussian noise during training together with other regularization methods and report that adding noise at a fixed noise level alone yields a noticeable increase in robustness. Contrary, Carlini and Wagner (2017) reevaluated the methods proposed by Zantedeschi et al. (2017) and reported that the robustness gains are small compared to adversarial training.

**Randomized smoothing.** Randomized smoothing is a very successful technique for obtaining robust classifiers (Cohen et al., 2019; Carlini et al., 2023), and is based on constructing a smoothed classifier from a base classifier by averaging the base classifier's outputs under Gaussian noise perturbation. The smoothed classifier is provably robust within a specified radii, without making any restrictions on the base classifier. Despite similarities at first sight, randomized smoothing considers surrogate smoothed models, whereas jittering is a training technique (see the appendix on a detailed discussion).

## 3  Theory for robust reconstruction of a signal lying in a subspace

In this section, we characterize the optimal robust estimator for denoising a signal in a subspace. While the resulting estimator is quite intuitive, proving optimality is fairly involved and relies on interesting applications of Jensen's inequality. We then show that the optimal robust estimator is also the unique minimizer of the jittering loss. Finally, we conjecture a precise characterization of optimal estimators for linear inverse problems beyond denoising, and show that jittering can result in suboptimal estimators for linear inverse problems beyond denoising.

### 3.1  Problem setup

We consider a signal reconstruction problem, where the goal is to estimate a signal $\mathbf{x}$ based on a noisy measurement $\mathbf{y} = \mathbf{A}\mathbf{x} + \mathbf{z}$, where $\mathbf{z} \sim \mathcal{N}(0, \sigma_z^2 1/m\mathbf{I})$ is Gaussian noise and $\mathbf{A} \in \mathbb{R}^{m \times n}$ a measurement or forward operator. The random noise is scaled so that the expected noise energy is $\mathbb{E}\left[\|\mathbf{z}\|_2^2\right] = \sigma_z^2$. The random noise is denoted by $\mathbf{z}$, to distinguish it from the adversarial noise or worst-case error, denoted by $\mathbf{e}$. We assume that the signal is (approximately) chosen uniformly from the intersection of a sphere and a subspace. Specifically, the signal is generated as $\mathbf{x} = \mathbf{U}\mathbf{c}$, where $\mathbf{c} \sim \mathcal{N}(0, \sigma_c^2 1/d\mathbf{I})$ is Gaussian and $\mathbf{U} \in \mathbb{R}^{n \times d}$ is an orthonormal basis for a $d$-dimensional subspace of $\mathbb{R}^n$. The expected signal energy is $\mathbb{E}\left[\|\mathbf{x}\|_2^2\right] = \sigma_c^2$.

We consider a linear estimator of the form $f(\mathbf{y}) = \mathbf{H}\mathbf{y}$ for estimating the signal from the measurement. For the standard reconstruction problem of estimating the signal $\mathbf{x}$ from the measurement $\mathbf{y}$, performance is often measured in terms of the expected least-squared error. We are interested in robust reconstruction and consider the expected worst-case reconstruction error with respect to an $\ell_2$-perturbation, defined in equation (1), and given by

$$R_\epsilon(f) = \mathbb{E}_{(\mathbf{x},\mathbf{y})}\left[\max_{\|\mathbf{e}\|_2 \leq \epsilon} \|\mathbf{H}(\mathbf{y} + \mathbf{e}) - \mathbf{x}\|_2^2\right].$$

For $\epsilon = 0$, the robust risk reduces to the standard expected mean-squared error.

### 3.2  Denoising

We start with denoising where the forward map is the identity, i.e., $\mathbf{A} = \mathbf{I}$. The following theorem characterizes the optimal worst-case robust denoiser.

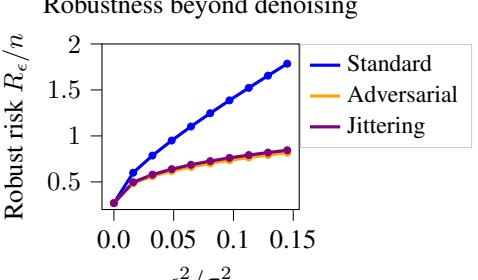

Figure 2: **Robust reconstruction of signals lying in a subspace.** The left panel shows the scaling factor $\alpha$ of the optimal robust denoiser according to Theorem 3.1 at noise levels $\sigma_z\sqrt{d/n} \in \{0, 0.4, 1.2\}$ and signal energy $\sigma_c^2 = 1$. The right panel depicts the robust risks of standard training and jittering as well as the optimal robust risk for an inverse problem beyond denoising, with the details stated in subsection 3.3.

**Theorem 3.1.** *For $d \rightarrow \infty$, the optimal worst-case estimator, i.e., the estimator minimizing the worst-case risk $R_\epsilon(f)$ amongst all estimators of the form $f(\mathbf{y}) = \mathbf{H}\mathbf{y}$, is $\mathbf{H} = \alpha\mathbf{U}\mathbf{U}^T$, where*

$$\alpha = \begin{cases} \dfrac{\sigma_c^2 - \dfrac{\epsilon\sigma_c\sigma_z\sqrt{\frac{d}{n}}}{\sqrt{\sigma_c^2 + \sigma_z^2\frac{d}{n} - \epsilon^2}}}{\sigma_c^2 + \sigma_z^2\frac{d}{n}} & \text{if } \sigma_c^2 > \epsilon^2 \\ 0 & \text{else.} \end{cases}$$

The worst-case optimal estimator projects onto the signal subspace, and then shrinks towards zero, by a factor determined by the noise variance $\sigma_z^2$ and the worst-case noise energy $\epsilon^2$. We consider the asymptotic setup where $d \rightarrow \infty$ only for expositional convenience; our proof shows that the estimator in the theorem is also near optimal for finite $d$.

To understand the implications of the theorem, let us first consider the case where the worst-case perturbation is zero. Then, the optimal estimator simply projects on the signal-subspace and shrinks towards zero, by a factor of $\alpha = \frac{\sigma_c^2}{\sigma_c^2 + \sigma_z^2\frac{d}{n}}$. The larger the noise, the more shrinkage.

Next, consider the most interesting regime, where non-zero adversarial noise is present. If the adversarial noise energy is larger than the signal energy, the estimator projects onto zero. However, this is an extreme regime since the adversarial noise can cancel the signal, and no good estimate of the signal can be achieved.

For the more practical regime where the adversarial noise energy is smaller than the signal energy, the theorem states that the optimal estimator projects onto the signal-subspace and shrinks towards zero—just like the optimal estimator for the noise-free case—but this time by a factor $\alpha$, that decreases in the adversarial noise energy $\epsilon^2$.

The proof of Theorem 3.1 is in the appendix. Note that for estimators $\mathbf{H} = \alpha\mathbf{U}\mathbf{U}^T$, a worst-case perturbation can be computed in closed form for a fixed $\mathbf{y}$ and $\mathbf{x}$: a worst-case perturbation is the vector that points into the direction of the signal plus noise lying in the signal subspace, i.e., $\mathbf{e} = \mathbf{U}\epsilon\frac{(1-\alpha)\mathbf{c}+\alpha\mathbf{U}^T\mathbf{z}}{\|(1-\alpha)\mathbf{c}+\alpha\mathbf{U}^T\mathbf{z}\|_2}$. However, for a general estimator $\mathbf{H}$, the perturbation can not be written in closed form, which makes proving optimality quite challenging. Our proof relies on a characterization of the inner maximization problem as the solution to an optimization problem in one variable, and several unusual applications of Jensen's inequality.

### 3.2.1 Robust denoisers via jittering

An important consequence of the characterization of the worst-case optimal estimator in Theorem 3.1 is that, at least for the linear setup considered in this section, a worst-case optimal estimator can be obtained by regularization with jittering.

Recall from the introduction that regularization via jittering simply adds Gaussian noise $\mathbf{w} \sim \mathcal{N}(0, \sigma_w^2 \mathbf{I})$ to the measurement during training. The jittering risk (2) of the estimator $f(\mathbf{y}) = \mathbf{H}\mathbf{y}$ for denoising is $J_{\sigma_w}(f) = \mathbb{E}_{(\mathbf{x},\mathbf{y}),\mathbf{w}}\left[\|f(\mathbf{y} + \mathbf{w}) - \mathbf{x}\|_2^2\right]$. Choosing the variance of the jittering noise level accordingly as a function of the desired robustness level $\epsilon$ yields an optimal worst-case robust estimator by minimizing the jittering loss, as formalized by the following corollary of Theorem 3.1.

**Corollary 3.2.** *For* $\epsilon^2 < \sigma_c^2$*, the linear estimator* $f(\mathbf{y}) = \mathbf{H}\mathbf{y}$ *that minimizes the jittering risk* $J_{\sigma_w}$ *with noise level chosen as a function of the desired noise level* $\epsilon$ *as* $\sigma_w(\epsilon) =$
$\sqrt{\dfrac{\epsilon^2 \sigma_z^2 \frac{d}{n} + \sigma_z \sqrt{\frac{d}{n}} \sigma_c \epsilon \sqrt{\sigma_c^2 - \epsilon^2 + \sigma_z^2 \frac{d}{n}}}{d(\sigma_c^2 - \epsilon^2)}}$ *also minimizes the worst-case risk* $R_\epsilon$.

Hence, if we aim for a robustness level $\epsilon < \sigma_c$, we can simply apply training via Jittering instead of adversarial training by choosing the Jittering noise level using the explicit formula for the jittering noise level $\sigma_w(\epsilon)$ in corollary 3.2.

Figure 1, left panel, shows the results of numerical simulation for adversarial training and jittering. In the implementation we treat the linear reconstructions as neural networks with a single layer without bias and perform adversarial training and jittering for each perturbation level. The simulations show that the robust risk performance of the models are identical, as predicted by the theory. Details on how adversarial training is performed are in Section 4.

### 3.2.2 Robustness accuracy trade-off

Another consequence of Theorem 3.1 is an explicit robustness-accuracy trade-off: increased worst-case robustness comes at a loss of accuracy. In the practically relevant regime of $0 \leq \epsilon^2 < \sigma_c^2$ the standard risk of the optimal worst-case estimator $f_{\alpha(\epsilon)}$ is $R_0(f_{\alpha(\epsilon)}) = \sigma_c^2 \cdot \dfrac{\sigma_z^2 \frac{d}{n}}{\sigma_c^2 + \sigma_z^2 \frac{d}{n} - \epsilon^2}$. This expression yields the optimal standard error for $\epsilon = 0$ and is strictly monotonically increasing in $\epsilon$, hence showing the loss of accuracy when increasing robustness. Robustness-accuracy tradeoffs can also be observed in other machine learning settings, for classification and regression settings, see for example (Tsipras et al., 2019). For linear inverse problems with applications in control, robustness accuracy-tradeoffs were recently characterized by Lee et al. (2021) and Javanmard et al. (2020).

### 3.3 General linear inverse problems

In the previous section, we characterized the optimal worst-case robust estimator and found that jittering yields optimal robust denoisers. In this section, we derive a conjecture for the worst-case optimal robust estimator for more general linear inverse problems of reconstructing a signal $\mathbf{x}$ from a measurement $\mathbf{y} = \mathbf{A}\mathbf{x} + \mathbf{z}$, with a forward operator $\mathbf{A}$ (with $\mathbf{A} \neq \mathbf{I}$ in general), and show that this estimator is in general not equal to the estimator obtained with jittering, thus jittering is in general suboptimal.

**Optimal robust estimator.** Let $\mathbf{A}\mathbf{U} = \mathbf{W}\mathbf{\Lambda}\mathbf{V}^T$ be the singular value decomposition of the matrix $\mathbf{A}\mathbf{U}$ with singular values $\lambda_i$. As formalized by Lemma A.1 of the appendix, the robust-risk (1) of the estimator $f$ can be written as an expectation involving a minimization problem over a single variable (instead of a maximization over an $n$-dimensional variable, as in the original definition):

$$R_\epsilon(\mathbf{H}) = \mathbb{E}_{\mathbf{v}}\left[\min_{\lambda \geq \sigma_{\max}^2} \lambda \epsilon^2 + \mathbf{v}^T(\mathbf{I} - \frac{1}{\lambda}\mathbf{H}\mathbf{H}^T)^{-1}\mathbf{v}\right], \quad \mathbf{v} = (\mathbf{H}\mathbf{A} - \mathbf{I})\mathbf{x} + \mathbf{H}\mathbf{z}. \quad (3)$$

Here, $\sigma_{\max}$ is the largest singular value of the matrix $\mathbf{H}$. In order to find the optimal robust estimator we wish to solve the optimization problem $\arg\min_{\mathbf{H}} R_\epsilon(\mathbf{H})$. The difficulty in solving this optimization problem is that we can't solve the minimization problem within the expectation (3) in closed form. In order to prove Theorem 3.1 for denoising (i.e., for $\mathbf{A} = \mathbf{I}$) we derived an upper and a matching lower bound of the risks using several unusual applications of Jensen's inequality. The proof does not generalize in a straightforward manner to the more general case where $\mathbf{A} \neq \mathbf{I}$. However, for large $d$, the random variable $\mathbf{v}^T(\mathbf{I} - \frac{1}{\lambda}\mathbf{H}\mathbf{H}^T)^{-1}\mathbf{v}$ concentrates around it's expectation, and thus we conjecture that for large $d$, we can exchange expectation and minimization, which yields:

$$R_\epsilon(\mathbf{H}) = \min_{\lambda \geq \sigma_{\max}^2} \lambda \epsilon^2 + \mathbb{E}_{\mathbf{v}}\left[\mathbf{v}^T(\mathbf{I} - \frac{1}{\lambda}\mathbf{H}\mathbf{H}^T)^{-1}\mathbf{v}\right]. \quad (4)$$

The expectation in the risk expression (4) can be explicitly computed, which yields the following conjecture for the worst-case optimal estimator:

**Conjecture 3.3.** For $d \to \infty$ the optimal worst-case estimator, i.e., the estimator minimizing the worst-case risk $R_\epsilon(f)$ amongst all estimators of the form $f(\mathbf{y}) = \mathbf{Hy}$, is $\mathbf{H} = \mathbf{UV}\mathrm{diag}(\sigma_i)\mathbf{W}^T$ with

$$\sigma_i = \frac{1 + \lambda\lambda_i^2}{2\lambda_i} + \frac{d}{m}\frac{\lambda}{2\lambda_i}\frac{\sigma_z^2}{\sigma_c^2} - \sqrt{\left(\frac{1 + \lambda\lambda_i^2}{2\lambda_i} + \frac{d}{m}\frac{\lambda}{2\lambda_i}\frac{\sigma_z^2}{\sigma_c^2}\right)^2 - \lambda},$$

if $\lambda_i \neq 0$ and $\sigma_i = 0$ otherwise. Here, the parameter $\lambda$ is a solution of:

$$\mathrm{argmin}_{\lambda \geq 0} \, \lambda\epsilon^2 + \sum_{i=1}^{d} \frac{1 - \lambda\lambda_i^2}{2}\frac{\sigma_c^2}{d} - \frac{\lambda}{2}\frac{\sigma_z^2}{m} + \sqrt{\left(\frac{1 + \lambda\lambda_i^2}{2}\frac{\sigma_c^2}{d} + \frac{\lambda}{2}\frac{\sigma_z^2}{m}\right)^2 - \lambda\lambda_i^2\frac{\sigma_c^4}{d^2}}.$$

The optimization problem involved is convex and box-constrained and can thus be solved numerically. Besides the argument above, we confirmed our conjecture with numerical simulations.

**Optimal jittering estimator.** Unlike for denoising, for general inverse problems, the jittering-risk minimizing estimator is in general not equal to the optimal worst-case estimator, but the two estimators are often close. The optimal estimator minimizing the jittering risk is given as (see Appendix C):

$$\mathbf{H}_J(\sigma_w) = \mathbf{UV}\mathrm{diag}\left(\frac{\sigma_c^2\lambda_i}{\sigma_c^2\lambda_i^2 + \sigma_z^2\frac{d}{m} + \sigma_w^2 d}\right)\mathbf{W}^T, \tag{5}$$

where as before $\mathbf{AU} = \mathbf{W\Lambda V}^T$ is the singular value decomposition with singular values $\lambda_i$. While the estimator (5) has the same form as the worst-case optimal estimator in Conjecture 3.3, the diagonal matrix in the two estimators is in general slightly different.

**Numerical Simulation.** The worst-case suboptimality of the jittering-risk optimal estimator (5) depends on the singular values of the matrix $\mathbf{AU}$; if they are equal the jittering-risk estimator is optimal, and if they are not equal there is typically a small gap. To illustrate the gap, we consider a forward operator $\mathbf{A}$ with linearly decaying singular values $\frac{i}{n}$, for $1 \leq i \leq n$ with signal energy $\sigma_c^2 = 1$ and noise level $\sigma_z = 0.2$. We compare the (conjectured) optimal robust estimator specified by Conjecture 3.3 with the optimal Jittering estimator (5) at noise level $\sigma_w$, where $\sigma_w$ is optimized such that one obtains minimal robust risk $R_\epsilon$ at a given perturbation level $\epsilon$. The results in Figure 2, right panel, show a small gap in robust risk, which implies that Jittering is suboptimal for this case. However, simulations with varying forward operators and noise levels indicate that the gap is small relative to the robust risk of the standard estimator. Experiments on image deconvolution using U-Net presented in Section 4 show similar results.

## 4 Experiments

In this section, we train standard convolutional neural-networks with standard training, adversarial training, and jittering for three inverse problems: denoising images, image deconvolution, and compressive sensing, and study their robustness. We find that Jittering yields well-performing robust denoisers at a computational cost similar to standard training, which is significantly cheaper than adversarial training. We also find that jittering yields robust estimators for deconvolution and compressive sensing. This indicates that training on real data which often contains slight measurement noise is robustness enhancing.

### 4.1 Problem setup

We start by describing the datasets, networks, and methodology.

**Natural images.** We consider denoising and deconvolution of natural images, where our goal is to reconstruct an image $\mathbf{x} \in \mathbb{R}^n$ from a noisy measurement $\mathbf{y} = \mathbf{Ax} + \mathbf{z}$, where $\mathbf{z} \sim \mathcal{N}(0, \sigma_z^2 1/n\mathbf{I})$ is Gaussian noise and $\mathbf{A}$ a measurement matrix, which is equal to identity for denoising, and implements

a convolution for deconvolution. For deconvolution we use a $8 \times 8$-sized discretization of the 2-dimensional Gaussian normal distribution with standard deviation 2. The kernel is visualized in Figure 3 in the appendix. We obtain train and validation datasets $\{(\mathbf{x}_1, \mathbf{y}_1), \ldots, (\mathbf{x}_N, \mathbf{y}_N)\}$ of sizes 34k and 4k, respectively, from colorized images of size $n = 128 \cdot 128 \cdot 3$ generated by randomly cropping and flipping ImageNet images. The methods are tested on 2k original-sized images.

**Medical data.** We also perform experiments on accelerated singlecoil magnetic resonance imaging (MRI) data, where the goal is to reconstruct an image $\mathbf{x} \in \mathbb{R}^n$ from a noisy and subsampled measurement in the frequency domain $\mathbf{y} = \mathbf{MFx} + \mathbf{z} \in \mathbb{R}^{2m}$. We use the fastMRI singlecoil knee dataset (Zbontar et al., 2018), which contains the images $\mathbf{x}$ and fully sampled measurements ($\mathbf{M} = \mathbf{I}$). We process it by random subsampling at acceleration factor 4 and obtain train, validation and test datasets with approximately 31k, 3.5k and 7k slices, respectively. While perturbations are sought in frequency domain, the inverse Fourier transform is applied to the measurements before feeding the $320 \times 320$ cropped and normalized images into the network.

**Network architecture.** We use the U-net architecture (Ronneberger et al., 2015) since it gives excellent performance for denoising (Brooks et al., 2019) and medical image reconstruction tasks, such as computed tomography (Jin et al., 2017) and is used as a building block for state-of-the-art methods for magnetic resonance imaging (Zbontar et al., 2018; Sriram et al., 2020). For natural images, we use a U-net with $3 \times 3$ padded convolutions with ReLU activation functions, $2 \times 2$ max-pooling layers for downscaling and transposed convolutions for upscaling. The network has 120k parameters. For MRI reconstruction we use a U-Net architecture similar to Zbontar et al. (2018) with $3 \times 3$ padded convolutions, leaky ReLU activation function, $2 \times 2$ average pooling and transposed convolutions (480k learnable parameters). We denote the U-Net by the parameterized mapping $f_{\boldsymbol{\theta}} \colon \mathbb{R}^m \to \mathbb{R}^n$ in the following.

**Evaluation.** We evaluate networks by measuring its robustness via the empirical robust risk defined as $\hat{R}_\epsilon(\boldsymbol{\theta}) = \frac{1}{N} \sum_{i=1}^{N} \max_{\|\mathbf{e}_i\|_2 \leq \epsilon} \|f_{\boldsymbol{\theta}}(\mathbf{y}_i + \mathbf{e}_i) - \mathbf{x}_i\|_2^2$, For evaluation, the robust empirical risk is computed over the test set. We assess the accuracy by computing the standard empirical risk $\hat{R}_0(\boldsymbol{\theta})$. Computing the robust empirical risk is non-trivial since it requires finding adversarial perturbations for solving the inner maximization problem. This is explained next. We also study the computational cost of the different methods, which we measure in terms of GPU cost and time.

**Finding adversarial perturbations.** To evaluate the empirical risk and for robust training, we need to compute adversarial perturbations $\mathbf{e} = \arg\max_{\|\mathbf{e}\|_2 \leq \epsilon} \|f_\theta(\mathbf{y} + \mathbf{e}) - \mathbf{x}\|_2^2$. We find the perturbations by running $N_a$ projected gradient ascent steps, starting with initial perturbation $\mathbf{e}^0 = 0$ and iterate

$$\mathbf{e}^{j+1} = \mathcal{P}_{B(0,\epsilon)}\left(\mathbf{e}^j + 2.5\frac{\epsilon}{N_a}\frac{\Delta\mathbf{e}^j}{\|\Delta\mathbf{e}^j\|_2}\right), \quad \text{where} \quad \Delta\mathbf{e}^j = \nabla_{\mathbf{e}^j}\|f_\theta(\mathbf{e}^j + \mathbf{y}) - \mathbf{x}\|_2^2.$$

Here, $\mathcal{P}_{B(0,\epsilon)}$ is the projection into the $\ell_2$-ball $B(0; \epsilon)$ of radius $\epsilon$ around the origin. The gradient is normalized to facilitate step size optimization with multiplier 2.5 such that the iteration can reach and move along the boundary, as suggested by Madry et al. (2018).

**Training methods.** **Standard training** minimizes the standard empirical risk $\hat{R}_0$. **Adversarial training** minimizes the empirical robust risk $\hat{R}_\epsilon$. To minimize the empirical robust risk, we approximate the inner maximization, $\max_{\|\mathbf{e}\|_2 \leq \epsilon} \|f_\theta(\mathbf{y}_i + \mathbf{e}) - \mathbf{x}_i\|_2^2$, with $\|f_\theta(\tilde{\mathbf{y}}_i) - \mathbf{x}_i\|_2^2$, where $\tilde{\mathbf{y}}_i$ is the adversarially perturbed measurement computed as described above. Training via jittering minimizes

$$\hat{J}_{\sigma_w}(\boldsymbol{\theta}) = \sum_{i=1}^{N} \mathbb{E}_{\mathbf{w} \sim \mathcal{N}(0, \sigma_w^2 \mathbf{I})}\left[\|f_\theta(\mathbf{y}_i + \mathbf{w}) - \mathbf{x}_i\|_2^2\right],$$

where the jittering level $\sigma_w$ is chosen depending on the desired robustness level. We treat the jittering noise level as a hyperparameter optimized using the validation dataset (shown in the appendix). Jittering is practically implemented via performing the SGD update rule $\boldsymbol{\theta}_{i+1} = \boldsymbol{\theta}_i - \frac{\eta}{M} \sum_{i=1}^{M} \nabla_{\boldsymbol{\theta}}\|f_\theta(\mathbf{y}_i + \mathbf{w}_i) - \mathbf{x}_i\|^2$, with $M$ samples per batch and learning rate $\eta$. For jittering, the network output is calculated on the noisy input $\mathbf{y}_i + \mathbf{w}_i$, instead of $\mathbf{y}_i$ (standard training) or

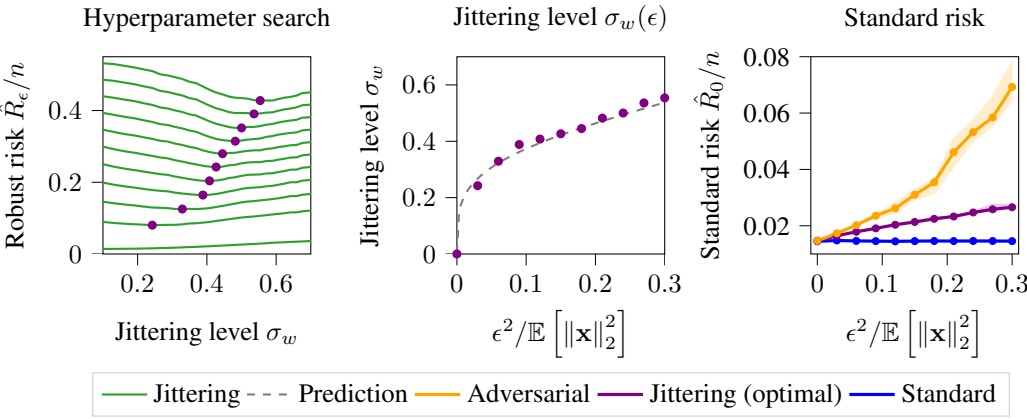

Figure 3: **Estimating the optimal jittering noise levels for the denoising task.** The left panel shows the results of training networks via Jittering, at noise levels $\sigma_w$, and calculating the empirical robust risk $\hat{R}_\epsilon$ of each model. Each green line corresponds to robust risks at one perturbation level. The optimal jittering noise levels are shown in the middle panel and follow well the prediction from theory (Cor. 3.2, details in appendix). The jittering estimators are similarly robust as adversarial training (Figure 1), but attain lower standard risks (right panel).

$\mathbf{y}_i + \mathbf{e}_i$ (adversarial training). To approximate the expectation in the jittering risk we draw independent jittering noise samples $\mathbf{w}_i$ in each iteration of SGD.

Throughout, we use PyTorch's Adam optimizer with learning rate $10^{-3}$ and batch size 50 for natural images, and $10^{-2}$ and 1 for MRI data. As perturbation levels, we consider values within the practically interesting regime of $\epsilon^2 / \mathbb{E}\left[\|\mathbf{A}\mathbf{x}\|_2^2\right] < 0.3$ for natural images and 0.03 for MRI data. Note that for $\epsilon^2 > \mathbb{E}\left[\|\mathbf{x}\|_2^2\right]$, Theorem 3.1 predicts for denoising ($\mathbf{A} = \mathbf{I}$) that the optimal robust estimator is zero everywhere. Figure 7 in the appendix shows that for large perturbations $\epsilon$ the trained U-net denoiser also maps to zero.

## 4.2 Results

We now discuss the results of the denoising, deconvolution, and compressive sensing experiments.

**Robust and standard performance.** Figure 1, shows that the standard estimator is relatively robust for Gaussian denoising and increasingly sensitive for more ill-posed problems (deconvolution and compressive sensing). The experiments further show that jittering is effective for enhancing robustness, in particular relative to the sensitivity of the standard estimator. Nevertheless, as suggested by theory, we see a gap between the robust risk of adversarial training and jittering for image deconvolution and compressive sensing. For Gaussian denoising, however, Jittering is particularly effective and yields increasingly better performing networks in terms of standard risks for larger perturbations.

**Choice of the jittering level.** The results are based on choosing the jittering noise levels via hyperparameter search for each task. Figure 3 shows the results for Gaussian denoising: It can be seen that the choice of noise level is important for minimizing the robust risk. The estimated noise levels also aligns well the theoretical prediction. Details on this and the parameter choices for deconvolution and the compressive sensing experiments are in the appendix.

**Computational complexity.** We measured the GPU time until convergence and memory utilization of the methods and present the results in the Table 1 of the appendix. Performing adversarial training is by a factor of the projected gradient ascent steps more expensive than standard training. Moreover, training via jittering has similar computational cost as standard training, since it solely consists of drawing and adding Gaussian noise on the training data.

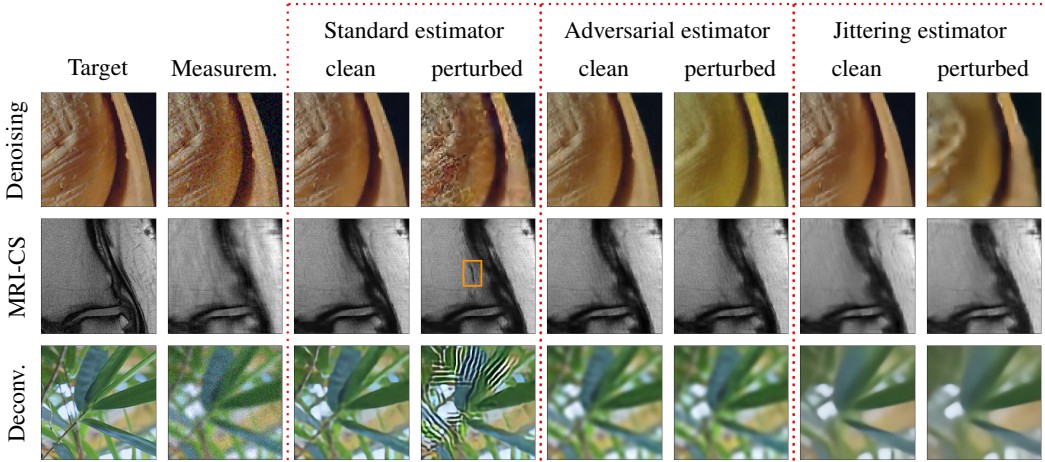

Figure 4: Example reconstructions using measurements (second column) and separately calculated perturbed measurements. The reconstructions are denoted as clean and perturbed, respectively. The perturbation levels are $\epsilon^2/\mathbb{E}[\|\mathbf{A}\mathbf{x}\|_2^2] = 0.03$ for denoising, 0.003 for compressive sensing and 0.001 for deconvolution. We can see that the standard estimator is visibly sensitive to perturbations. Jittering yields robust estimators, but at the same time yields smoother reconstructions.

**Visual reconstructions.** For the linear subspace setting adversarial training and jittering are equivalent. For Gaussian denoising with a neural network, however, they perform differently. For larger perturbations jittering tends to yield smoother images than networks trained adversarially, as can be seen in the example reconstructions shown in Figure 4. This effect is particularly noticeable for the Gaussian deconvolution task. We quantified the effect using the total variation norm and present the results in the appendix. Moreover, we discuss an approximation to jittering, Jacobian regularization, which similarly enhances robustness. It is computationally more expensive, but yields less smooth reconstructions.

## 5 Conclusion

In this paper, we characterized the optimal worst-case robust estimator for Gaussian subspace denoising and found that the optimal estimator can be provably learned with jittering. Our results for training neural networks for Gaussian denoising of images show that jittering enables the training of neural networks that are as robust as neural networks trained adversarially, but at a fraction of the computational cost, and without the hassle of having to find adversarial perturbations. While we demonstrated that jittering can yield suboptimal robust estimators in general, in practice, jittering is effective at improving the robustness for compressive sensing and image deconvolution. Moreover, our results imply that training on real data that contains slight measurement noise is robustness enhancing.

**Reproducability** The repository at `https://github.com/MLI-lab/robust_reconstructors_via_jittering` contains the code to reproduce all results in the main body of this paper.

**Acknowledgments** A.K. and R.H. are supported by the Institute of Advanced Studies at the Technical University of Munich, the Deutsche Forschungsgemeinschaft (DFG, German Research Foundation) - 456465471, 464123524, the DAAD, and the German Federal Ministry of Education and Research, and the Bavarian State Ministry for Science and the Arts. M.S. is supported by the Packard Fellowship in Science and Engineering, a Sloan Research Fellowship in Mathematics, an NSF-CAREER under award #1846369, DARPA FastNICS programs, and NSF-CIF awards #1813877 and #2008443.

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

# A    Proof of Theorem 3.1

In the main body we stated an analytical characterization of the optimal worst-case robust denoiser. We present the proof in the following and show that the risk

$$\min_{\mathbf{H}} R_\epsilon(\mathbf{H}) = \min_{\mathbf{H}} \mathbb{E}_{\mathbf{x},\mathbf{z}} \left[ \max_{\|\mathbf{e}\|_2 \le \epsilon} \|\mathbf{H}(\mathbf{x} + \mathbf{e} + \mathbf{z}) - \mathbf{x}\|_2^2 \right] \tag{6}$$

is minimized by $\mathbf{H} = \sigma \mathbf{U} \mathbf{U}^T$ with

$$\sigma = \begin{cases} \dfrac{\sigma_c^2 - \dfrac{\epsilon \sigma_c \sigma_z \sqrt{\frac{d}{n}}}{\sqrt{\sigma_c^2 + \sigma_z^2 \frac{d}{n} - \epsilon^2}}}{\sigma_c^2 + \sigma_z^2 \frac{d}{n}} & \text{if } \sigma_c^2 > \epsilon^2 \\ 0 & \text{else.} \end{cases}$$

The scaling factor $\sigma$ as a function of the noise level $\epsilon$ of the optimal worst-case estimator is shown in Figure 2.

We start by proving a lower bound of the risk, which relies on a characterization of the maximization and many unexpected applications of Jensens inequality.

We then compute the risk for $\mathbf{H} = \sigma \mathbf{U} \mathbf{U}^T$ and show that it is equivalent to the lower bound on the risk.

## A.1    Lower bounding the risk

Towards lower bounding the risk we define for notational convenience

$$\begin{bmatrix} \mathbf{H}_\| & \mathbf{H}_0 \\ \mathbf{H}_1 & \mathbf{H}_\perp \end{bmatrix} = \begin{bmatrix} \mathbf{U}^T \mathbf{H} \mathbf{U} & \mathbf{U}^T \mathbf{H} \mathbf{U}_\perp \\ \mathbf{U}_\perp^T \mathbf{H} \mathbf{U} & \mathbf{U}_\perp^T \mathbf{H} \mathbf{U}_\perp \end{bmatrix}.$$

With

$$\mathbf{H} = [\mathbf{U} \quad \mathbf{U}_\perp] \begin{bmatrix} \mathbf{H}_\| & \mathbf{H}_0 \\ \mathbf{H}_1 & \mathbf{H}_\perp \end{bmatrix} \begin{bmatrix} \mathbf{U}^T \\ \mathbf{U}_\perp^T \end{bmatrix}$$

we get

$$\max_{\|\mathbf{e}\|_2 \le \epsilon} \|\mathbf{H}(\mathbf{x} + \mathbf{e} + \mathbf{z}) - \mathbf{x}\|_2^2 = \max_{\|\mathbf{e}\|_2 \le \epsilon} \left\| [\mathbf{U} \quad \mathbf{U}_\perp] \begin{bmatrix} \mathbf{H}_\| & \mathbf{H}_0 \\ \mathbf{H}_1 & \mathbf{H}_\perp \end{bmatrix} \begin{bmatrix} \mathbf{U}^T \\ \mathbf{U}_\perp^T \end{bmatrix} (\mathbf{U}\mathbf{c} + \mathbf{e} + \mathbf{z}) - \mathbf{U}\mathbf{c} \right\|_2^2$$

$$= \max_{\|\mathbf{e}\|_2 \le \epsilon} \left\| \begin{bmatrix} \mathbf{H}_\| & \mathbf{H}_0 \\ \mathbf{H}_1 & \mathbf{H}_\perp \end{bmatrix} \begin{bmatrix} \mathbf{c} + \mathbf{U}^T\mathbf{z} + \mathbf{U}^T\mathbf{e} \\ \mathbf{U}_\perp^T\mathbf{z} + \mathbf{U}_\perp^T\mathbf{e} \end{bmatrix} - \begin{bmatrix} \mathbf{c} \\ 0 \end{bmatrix} \right\|_2^2$$

$$\ge \max_{\|\mathbf{e}\|_2 \le \epsilon} \left\| [\mathbf{H}_\| \quad \mathbf{H}_0] \begin{bmatrix} \mathbf{c} + \mathbf{U}^T\mathbf{z} + \mathbf{U}^T\mathbf{e} \\ \mathbf{U}_\perp^T\mathbf{z} + \mathbf{U}_\perp^T\mathbf{e} \end{bmatrix} - \mathbf{c} \right\|_2^2$$

$$\ge \max_{\|\mathbf{e}_\|\|_2 \le \epsilon} \left\| \mathbf{H}_\|(\mathbf{c} + \mathbf{z}_\| + \mathbf{e}_\|) - \mathbf{c} + \mathbf{H}_0 \mathbf{z}_\perp \right\|_2^2.$$

Here, we defined $\mathbf{e}_\| = \mathbf{U}^T \mathbf{e}$ and $\mathbf{z}_\| = \mathbf{U}^T \mathbf{z}$ for notational convenience, and the last inequality follows by adding $\mathbf{U}_\perp^T \mathbf{e} = 0$ as a constraint to the maximization, which gives a lower bound.

Next, note that since $\mathbf{z} = \begin{bmatrix} \mathbf{z}_\| \\ \mathbf{z}_\perp \end{bmatrix} \sim \mathcal{N}(0, \sigma_z^2/n \mathbf{I})$ is Gaussian, the vector $\begin{bmatrix} \mathbf{z}_\| \\ -\mathbf{z}_\perp \end{bmatrix}$ is equally Gaussian distributed $\mathcal{N}(0, \sigma_z^2/n \mathbf{I})$. This implies that

$$\mathbb{E}\left[ \max_{\|\mathbf{e}_\|\|_2 \le \epsilon} \left\| \mathbf{H}_\|(\mathbf{c} + \mathbf{z}_\| + \mathbf{e}_\|) - \mathbf{c} + \mathbf{H}_0 \mathbf{z}_\perp \right\|_2^2 \right] = \mathbb{E}\left[ \max_{\|\mathbf{e}_\|\|_2 \le \epsilon} \left\| \mathbf{H}_\|(\mathbf{c} + \mathbf{z}_\| + \mathbf{e}_\|) - \mathbf{c} - \mathbf{H}_0 \mathbf{z}_\perp \right\|_2^2 \right].$$

Then

$$\mathbb{E}\left[\max_{\|\mathbf{e}_\|\|_2\leq\epsilon}\left\|\mathbf{H}_\|(\mathbf{c}+\mathbf{z}_\|+\mathbf{e}_\|)-\mathbf{c}+\mathbf{H}_0\mathbf{z}_\perp\right\|_2^2\right]$$

$$=\frac{1}{2}\mathbb{E}\left[\max_{\|\mathbf{e}_1\|_2\leq\epsilon}\left\|\mathbf{H}_\|(\mathbf{c}+\mathbf{z}_\|+\mathbf{e}_1)-\mathbf{c}+\mathbf{H}_0\mathbf{z}_\perp\right\|_2^2+\max_{\|\mathbf{e}_2\|_2\leq\epsilon}\left\|\mathbf{H}_\|(\mathbf{c}+\mathbf{z}_\|+\mathbf{e}_2)-\mathbf{c}-\mathbf{H}_0\mathbf{z}_\perp\right\|_2^2\right]$$

$$\geq\mathbb{E}\left[\max_{\|\mathbf{e}_1\|_2\leq\epsilon}\frac{1}{2}\left\|\mathbf{H}_\|(\mathbf{c}+\mathbf{z}_\|+\mathbf{e}_1)-\mathbf{c}+\mathbf{H}_0\mathbf{z}_\perp\right\|_2^2+\frac{1}{2}\left\|\mathbf{H}_\|(\mathbf{c}+\mathbf{z}_\|+\mathbf{e}_1)-\mathbf{c}-\mathbf{H}_0\mathbf{z}_\perp\right\|_2^2\right]$$

$$\geq\mathbb{E}\left[\max_{\|\mathbf{e}_1\|_2\leq\epsilon}\left\|\mathbf{H}_\|(\mathbf{c}+\mathbf{z}_\|+\mathbf{e}_1)-\mathbf{c}\right\|_2^2\right],$$

where the last step follows from Jensens inequality. Thus, we have shown that

$$\min_{\mathbf{H}}\mathbb{E}\left[\max_{\|\mathbf{e}\|_2\leq\epsilon}\left\|\mathbf{H}(\mathbf{x}+\mathbf{e}+\mathbf{z})-\mathbf{x}\right\|_2^2\right]\geq\min_{\mathbf{H}_\|}\mathbb{E}\left[\max_{\|\mathbf{e}\|_2\leq\epsilon}\left\|\mathbf{H}_\|(\mathbf{c}+\mathbf{z}+\mathbf{e})-\mathbf{c}\right\|_2^2\right].\quad(7)$$

For simplicity of exposition, we drop the $\|$-notation. Thus, with a slight abuse of notation, the vectors on the left and right hand side have different dimensions. On the left hand side, $\mathbf{z},\mathbf{e}\in\mathbb{R}^n$, while on the right hand side $\mathbf{z},\mathbf{e}\in\mathbb{R}^d$ and $\mathbf{z}$ throughout has iid $\mathcal{N}(0,\sigma_z^2/n)$ entries.

Next, we'll apply the lemma below for characterizing the maximization inside of the expectation. The proof is in Section A.3. Similar computations as used to prove the lemma are on page 19-20 in the paper Lee et al. (2021) for deriving robustness-accuracy trade-off bounds.

**Lemma A.1.** *For any* $\mathbf{z}$,

$$\max_{\|\mathbf{e}\|_2\leq\epsilon}\|\mathbf{z}-\mathbf{H}\mathbf{e}\|_2^2=\min_{\lambda:\,\lambda\geq\lambda_{\max}(\mathbf{H}^T\mathbf{H})}\lambda\epsilon^2+\mathbf{z}^T\left(\mathbf{I}-\frac{1}{\lambda}\mathbf{H}\mathbf{H}^T\right)^{-1}\mathbf{z}.\quad(8)$$

Using Lemma A.1 the term within the expectation is

$$\max_{\|\mathbf{e}\|_2\leq\epsilon}\left\|(\mathbf{H}-\mathbf{I})\mathbf{c}+\mathbf{H}\mathbf{z}+\mathbf{H}\mathbf{e}\right\|_2^2$$

$$=\min_{\lambda:\,\lambda\geq\lambda_{\max}(\mathbf{H}^T\mathbf{H})}\lambda\epsilon^2+((\mathbf{H}-\mathbf{I})\mathbf{c}+\mathbf{H}\mathbf{z})^T\left(\mathbf{I}-\frac{1}{\lambda}\mathbf{H}\mathbf{H}^T\right)^{-1}((\mathbf{H}-\mathbf{I})\mathbf{c}+\mathbf{H}\mathbf{z})$$

$$=\min_{\lambda:\,\lambda\geq\lambda_{\max}(\mathbf{H}^T\mathbf{H})}\lambda\epsilon^2+\mathbf{c}^T(\mathbf{H}-\mathbf{I})^T\left(\mathbf{I}-\frac{1}{\lambda}\mathbf{H}\mathbf{H}^T\right)^{-1}(\mathbf{H}-\mathbf{I})\mathbf{c}$$

$$+2\mathbf{z}^T\mathbf{H}^T\left(\mathbf{I}-\frac{1}{\lambda}\mathbf{H}\mathbf{H}^T\right)^{-1}(\mathbf{H}-\mathbf{I})\mathbf{c}$$

$$+\mathbf{z}^T\mathbf{H}^T\left(\mathbf{I}-\frac{1}{\lambda}\mathbf{H}\mathbf{H}^T\right)^{-1}\mathbf{H}\mathbf{z}.$$

Now let's rewrite using the singular value decomposition $\mathbf{H}_\|=\mathbf{V}\boldsymbol{\Sigma}\mathbf{W}^T$, with orthogonal matrices $\mathbf{V}$ and $\mathbf{W}$. With this notation, the terms above can be rewritten as

$$\mathbf{c}^T(\mathbf{V}\boldsymbol{\Sigma}\mathbf{W}^T-\mathbf{I})^T\left(\mathbf{I}-\frac{1}{\lambda}\mathbf{V}\boldsymbol{\Sigma}\mathbf{W}^T(\mathbf{V}\boldsymbol{\Sigma}\mathbf{W}^T)^T\right)^{-1}(\mathbf{V}\boldsymbol{\Sigma}\mathbf{W}^T-\mathbf{I})\mathbf{c}$$

$$=\mathbf{c}^T(\mathbf{W}\boldsymbol{\Sigma}\mathbf{V}^T-\mathbf{I})\mathbf{V}\left(\mathbf{I}-\frac{1}{\lambda}\boldsymbol{\Sigma}^2\right)^{-1}\mathbf{V}^T(\mathbf{V}\boldsymbol{\Sigma}\mathbf{W}^T-\mathbf{I})\mathbf{c}$$

$$=\mathbf{c}^T(\mathbf{W}\boldsymbol{\Sigma}-\mathbf{V})\left(\mathbf{I}-\frac{1}{\lambda}\boldsymbol{\Sigma}^2\right)^{-1}(\boldsymbol{\Sigma}\mathbf{W}^T-\mathbf{V}^T)\mathbf{c}=\sum_{i=1}^d\frac{\left((\sigma_i\mathbf{w}_i-\mathbf{v}_i)^T\mathbf{c}\right)^2}{1-\frac{\sigma_i^2}{\lambda}},$$

where $\mathbf{w}_i$ and $\mathbf{v}_i$ are the $i$-th column vectors of the orthogonal matrices $\mathbf{W}$ and $\mathbf{V}$, respectively. Using similar calculations for the other summands, the optimization problem on the right hand side of inequality (7) is

$$\min_{\mathbf{H}_\parallel} \mathbb{E}\left[\max_{\|\mathbf{e}\|_2 \le \epsilon} \left\|\mathbf{H}_\parallel(\mathbf{c}+\mathbf{z}+\mathbf{e})-\mathbf{c}\right\|_2^2\right]$$

$$= \min_{\mathbf{V},\mathbf{W},\sigma_i} \mathbb{E}_{\mathbf{c},\mathbf{z}}\left[\min_{\lambda:\,\lambda\ge\sigma_i^2} \lambda\epsilon^2 + \sum_{i=1}^d \frac{\left((\sigma_i\mathbf{w}_i-\mathbf{v}_i)^T\mathbf{c}\right)^2 + 2\left((\sigma_i\mathbf{w}_i-\mathbf{v}_i)^T\mathbf{c}\right)\left(\sigma_i\mathbf{w}_i^T\mathbf{z}\right) + \left(\sigma_i\mathbf{w}_i^T\mathbf{z}\right)^2}{1-\frac{\sigma_i^2}{\lambda}}\right]$$

$$= \min_{\mathbf{V},\mathbf{W},\sigma_i} \mathbb{E}_{\mathbf{c},\mathbf{z}}\left[\min_{\lambda:\,\lambda\ge\sigma_i^2} \lambda\epsilon^2 + \sum_{i=1}^d \frac{\left((\sigma_i\mathbf{w}_i-\mathbf{v}_i)^T\mathbf{c}+\sigma_i\mathbf{w}_i^T\mathbf{z}\right)^2}{1-\frac{\sigma_i^2}{\lambda}}\right]$$

$$\overset{\text{i}}{=} \min_{\mathbf{V},\mathbf{W},\sigma_i} \mathbb{E}_{\mathbf{g}}\left[\min_{\lambda:\,\lambda\ge\sigma_i^2} \lambda\epsilon^2 + \sum_{i=1}^d \frac{g_i^2\left((\sigma_i^2+1-2\sigma_i\mathbf{v}_i{}^T\mathbf{w}_i)\frac{\sigma_c^2}{d}+\sigma_i^2\frac{\sigma_z^2}{n}\right)}{1-\frac{\sigma_i^2}{\lambda}}\right]$$

$$\overset{\text{ii}}{\ge} \min_{\mathbf{V},\mathbf{W},\sigma_i} \mathbb{E}_{\mathbf{g}}\left[\min_{\lambda:\,\lambda\ge\sigma_i^2} \lambda\epsilon^2 + \sum_{i=1}^d \frac{g_i^2\left((\sigma_i^2+1-2\sigma_i)\frac{\sigma_c^2}{d}+\sigma_i^2\frac{\sigma_z^2}{n}\right)}{1-\frac{\sigma_i^2}{\lambda}}\right]$$

$$= \min_{\sigma_i} \mathbb{E}_{\mathbf{g}}\left[\min_{\lambda:\,\lambda\ge\sigma_i^2} \lambda\epsilon^2 + \sum_{i=1}^d \frac{g_i^2\left(\frac{\sigma_c^2}{d}(\sigma_i-1)^2+\frac{\sigma_z^2}{n}\sigma_i^2\right)}{1-\frac{\sigma_i^2}{\lambda}}\right],$$

where minimization above is over the orthonormal matrices $\mathbf{V},\mathbf{W} \in \mathbb{R}^{d\times d}$ and over the singular values $\sigma_i$. Equality (i) follows by noting that $(\sigma_i\mathbf{w}_i^T-\mathbf{v}_i{}^T)\mathbf{c}$ is Gaussian with variance $(\sigma_i^2+1-2\sigma_i(\mathbf{v}_i{}^T\mathbf{w}_i))\frac{\sigma_c^2}{d}$ and $(\sigma_i\mathbf{w}_i^T)\mathbf{z}$ is Gaussian with variance $\sigma_i^2\frac{\sigma_z^2}{n}$ and therefore we can replace the expectation with respect to $(\mathbf{c},\mathbf{z})$ with the expectation with respect to $\mathbf{g}\sim\mathcal{N}(0,\mathbf{I})$ and suitable scaling. Moreover, inequality (ii) follows from the fact that $\sigma_i\ge 0$ and $g_i^2\ge 0$.

Continuing, first note that the function $\frac{x^2}{y}$ is convex in $(x,y)$ when $y>0$. Also the extended value function is increasing in the first input and decreasing in the second input. Furthermore the mappings $(x,z)\mapsto x-1$ and $(x,z)\mapsto 1-\frac{x^2}{z}$ are convex and concave. Thus by the composition rule of convex functions we conclude that the functions

$$\frac{x^2}{1-\frac{(x-1)^2}{z}} \quad \text{and} \quad \frac{(x-1)^2}{1-\frac{(x-1)^2}{z}}$$

are jointly convex in $(x,z)$.

Jensen's inequality states that for a convex function $\psi$ we have $\frac{\sum_i z_i^2\psi(x_i)}{\sum_i z_i^2}\ge\psi\left(\frac{\sum_i z_i^2\mathbf{x}_i}{\sum_i z_i^2}\right)$. Thus by Jensen's inequality the sum in the expectation in the right-hand-side of the equation above can be lower-bounded as

$$\sum_{i=1}^d \frac{g_i^2\frac{\sigma_c^2}{d}(\sigma_i-1)^2+g_i^2\frac{\sigma_z^2}{n}\sigma_i^2}{1-\frac{\sigma_i^2}{\lambda}} = \|\mathbf{g}\|_2^2\sum_{i=1}^d \frac{\frac{g_i^2}{\|\mathbf{g}\|_2^2}\frac{\sigma_c^2}{d}(\sigma_i-1)^2+\frac{g_i^2}{\|\mathbf{g}\|_2^2}\frac{\sigma_z^2}{n}\sigma_i^2}{1-\frac{\sigma_i^2}{\lambda}}$$

$$\ge \|\mathbf{g}\|_2^2\frac{\frac{\sigma_c^2}{d}\left(\sum_{i=1}^d\frac{g_i^2}{\|\mathbf{g}\|_2^2}\sigma_i-1\right)^2+\frac{\sigma_z^2}{n}\left(\sum_{i=1}^d\frac{g_i^2}{\|\mathbf{g}\|_2^2}\sigma_i\right)^2}{1-\frac{\left(\sum_{i=1}^d\frac{g_i^2}{\|\mathbf{g}\|_2^2}\sigma_i\right)^2}{\lambda}}$$

$$= \|\mathbf{g}\|_2^2\frac{\frac{\sigma_c^2}{d}\left(\bar{\sigma}(\mathbf{g})-1\right)^2+\frac{\sigma_z^2}{n}\left(\bar{\sigma}(\mathbf{g})\right)^2}{1-\frac{(\bar{\sigma}(\mathbf{g}))^2}{\lambda}}$$

where $\bar{\sigma}(\mathbf{g})=\sum_{i=1}^d\frac{g_i^2}{\|\mathbf{g}\|_2^2}\sigma_i$.

Now consider the event $\mathcal{E} = \{\mathbf{g}\colon \|\mathbf{g}\|_2^2 \geq (1-\delta)d\}$ which holds with probability at least $1 - e^{-\frac{\delta^2}{2}d}$. On this event, we have

$$\lambda\epsilon^2 + \sum_{i=1}^{d} \frac{g_i^2 \frac{\sigma_c^2}{d}(\sigma_i - 1)^2 + g_i^2 \frac{\sigma_z^2}{n}\sigma_i^2}{1 - \frac{\sigma_i^2}{\lambda}} \geq \lambda\epsilon^2 + (1-\delta)\frac{\sigma_c^2 (\bar\sigma(\mathbf{g}) - 1)^2 + \sigma_z^2 \frac{d}{n}(\bar\sigma(\mathbf{g}))^2}{1 - \frac{(\bar\sigma(\mathbf{g}))^2}{\lambda}}.$$

Using the same argument as before the right hand side of the above inequality is jointly convex in $(\lambda, \bar\sigma(\mathbf{c}))$. Since partial minimization of a jointly convex function preserves convexity we conclude that the function

$$\psi(\bar\sigma) = \min_{\lambda\colon \lambda \geq \sigma_i^2} \mathbb{1}_{\mathcal{E}}\lambda\epsilon^2 + \mathbb{1}_{\mathcal{E}}(1-\delta)\frac{\sigma_c^2 (\bar\sigma(\mathbf{g}) - 1)^2 + \sigma_z^2 \frac{d}{n}(\bar\sigma(\mathbf{g}))^2}{1 - \frac{(\bar\sigma(\mathbf{g}))^2}{\lambda}}$$

is convex in $\bar\sigma$. Thus by using convexity in terms of $\bar\sigma$, applying Jensen's inequality (i.e., $\mathbb{E}\left[\psi(\bar\sigma)\right] \geq \psi(\mathbb{E}\left[\bar\sigma\right])$) we have

$$\min_{\sigma_i} \mathbb{E}_{\mathbf{g}}\left[\min_{\lambda\colon \lambda \geq \sigma_i^2} \lambda\epsilon^2 + \|\mathbf{g}\|_2^2 \frac{1}{d}\frac{\sigma_c^2(\bar\sigma(\mathbf{g})-1)^2 + \sigma_z^2\frac{d}{n}(\bar\sigma(\mathbf{g}))^2}{1 - \frac{(\bar\sigma(\mathbf{g}))^2}{\lambda}}\right]$$

$$\geq \min_{\sigma_i} \mathbb{E}_{\mathbf{g}}\left[\min_{\lambda\colon \lambda \geq \sigma_i^2} \mathbb{1}_{\mathcal{E}}\lambda\epsilon^2 + \mathbb{1}_{\mathcal{E}}(1-\delta)\frac{\sigma_c^2(\bar\sigma(\mathbf{g})-1)^2 + \sigma_z^2\frac{d}{n}(\bar\sigma(\mathbf{g}))^2}{1 - \frac{(\bar\sigma(\mathbf{g}))^2}{\lambda}}\right]$$

$$\geq \min_{\sigma_i} \mathbb{E}_{\mathbf{g}|\mathcal{E}}\left[\min_{\lambda\colon \lambda \geq \sigma_i^2} \lambda\epsilon^2 + (1-\delta)\frac{\sigma_c^2(\bar\sigma(\mathbf{g})-1)^2 + \sigma_z^2\frac{d}{n}(\bar\sigma(\mathbf{g}))^2}{1 - \frac{(\bar\sigma(\mathbf{g}))^2}{\lambda}}\right]$$

$$\geq \min_{\sigma_i} \min_{\lambda\colon \lambda \geq \sigma_i^2} \lambda\epsilon^2 + (1-\delta)\frac{\sigma_c^2(\mathbb{E}\left[\bar\sigma(\mathbf{g})\right]-1)^2 + \sigma_z^2\frac{d}{n}(\mathbb{E}\left[\bar\sigma(\mathbf{g})\right])^2}{1 - \frac{(\mathbb{E}[\bar\sigma(\mathbf{g})])^2}{\lambda}}$$

$$\geq \min_{\sigma_i} \min_{\lambda\colon \lambda \geq \sigma_i^2} \lambda\epsilon^2 + (1-\delta)\frac{\sigma_c^2(\bar\sigma-1)^2 + \sigma_z^2\frac{d}{n}(\bar\sigma)^2}{1 - \frac{(\bar\sigma)^2}{\lambda}}$$

where we defined $\bar\sigma = \sum_{i=1}^{d} \mathbb{E}_{\mathbf{g}|\mathcal{E}}\left[\frac{g_i^2}{\|\mathbf{g}\|_2^2}\right]\sigma_i$. Putting things together, we have shown that

$$\min_{\mathbf{H}} R_\epsilon(\mathbf{H}) \geq \min_{\mathbf{H}_\|} \mathbb{E}\left[\max_{\|\mathbf{e}\|_2 \leq \epsilon} \left\|\mathbf{H}_\|(\mathbf{c} + \mathbf{z} + \mathbf{e}) - \mathbf{c}\right\|_2^2\right]$$

$$= \min_{\sigma_i} \mathbb{E}_{\mathbf{g}}\left[\min_{\lambda\colon \lambda \geq \sigma_i^2} \lambda\epsilon^2 + \sum_{i=1}^{d} \frac{g_i^2 \frac{\sigma_c^2}{d}(\sigma_i - 1)^2 + g_i^2 \frac{\sigma_z^2}{n}\sigma_i^2}{1 - \frac{\sigma_i^2}{\lambda}}\right]$$

$$\geq \min_{\bar\sigma} \min_{\lambda\colon \lambda \geq \sigma_i^2} \lambda\epsilon^2 + (1-\delta)\frac{\sigma_c^2(\bar\sigma-1)^2 + \sigma_z^2\frac{d}{n}(\bar\sigma)^2}{1 - \frac{(\bar\sigma)^2}{\lambda}}$$

$$\geq \min_{\bar\sigma} \min_{\lambda\colon \lambda \geq \bar\sigma^2} \lambda\epsilon^2 + (1-\delta)\frac{\sigma_c^2(\bar\sigma-1)^2 + \sigma_z^2\frac{d}{n}(\bar\sigma)^2}{1 - \frac{(\bar\sigma)^2}{\lambda}},$$

where the last inequality follows from $\sigma_{\max} \geq \bar\sigma$. For $d \to \infty$, we can choose $\delta$ arbitrarily small, which yields

$$\min_{\mathbf{H}} R_\epsilon(\mathbf{H}) \geq \min_{\sigma} \min_{\lambda\colon \lambda \geq \sigma^2} \lambda\epsilon^2 + \frac{\sigma_c^2(\sigma - 1)^2 + \sigma_z^2\frac{d}{n}\sigma^2}{1 - \frac{\sigma^2}{\lambda}}. \tag{9}$$

**Solving the optimization problem** (9): Consider the inner minimization problem in equation (9), i.e.,

$$\min_{\lambda\colon \lambda \geq \sigma^2} f(\lambda) \qquad \text{with } f(\lambda) = \lambda\epsilon^2 + \frac{c(\sigma)}{1 - \frac{\sigma^2}{\lambda}},$$

where $c(\sigma) = \sigma_c^2 (\sigma - 1)^2 + \sigma_z^2 \frac{d}{n} \sigma^2$ for notational convenience. Since $f$ is differentiable on $(\sigma^2, \infty)$ we can calculate its critical point $\lambda^*$ by setting the derivative with respect to $\lambda$ to zero, which yields

$$\lambda^* = \frac{\sqrt{c(\sigma)}}{\epsilon} \sigma + \sigma^2.$$

From this expression, we see that the constraint $\lambda^* > \sigma^2$ is satisfied and by convexity of $f$ we know that $\lambda^*$ is the unique minimizer. Hence, we have:

$$\min_{\lambda : \lambda \geq \sigma^2} f(\lambda) = f(\lambda^*) = \left( \epsilon\sigma + \sqrt{c(\sigma)} \right)^2. \tag{10}$$

It follows that

$$\min_{\mathbf{H}} R_\epsilon(\mathbf{H}) \geq \min_\sigma g(\sigma), \quad g(\sigma) = \left( \epsilon\sigma + \sqrt{\sigma_c^2(\sigma - 1)^2 + \sigma_z^2 \frac{d}{n} \sigma^2} \right)^2. \tag{11}$$

Calculating the derivative of $g(\sigma)$ and setting it to zero, we get:

$$2 \left( \epsilon\sigma + \sqrt{\sigma_c^2(\sigma - 1)^2 + \sigma_z^2 \frac{d}{n} \sigma^2} \right) \left( \epsilon + \frac{2\sigma_c^2(\sigma - 1) + \sigma_z^2 \frac{d}{n} 2\sigma}{2\sqrt{\sigma_c^2(\sigma - 1)^2 + \sigma_z^2 \frac{d}{n} \sigma^2}} \right) = 0$$

The left factor is non-negative, and the right factor is zero if $\sigma_c^2 > \epsilon^2$ and if

$$\sigma^* = \frac{\sigma_c^2 - \frac{\epsilon \sigma_c \sigma_z \sqrt{\frac{d}{n}}}{\sqrt{\sigma_c^2 + \sigma_z^2 \frac{d}{n} - \epsilon^2}}}{\sigma_c^2 + \sigma_z^2 \frac{d}{n}}. \tag{12}$$

If $\sigma_c^2 \leq \epsilon^2$, then the function $g(\sigma)$ is monotonically increasing on $[0, \infty)$ and hence $\sigma_* = 0$ is the minimizer.

### A.2 Upper bound for the risk of the estimator $\mathbf{H} = \sigma\mathbf{U}\mathbf{U}^T$

We upper bound the risk of the estimator $\mathbf{H} = \sigma\mathbf{U}\mathbf{U}^T$. For $\mathbf{H} = \sigma\mathbf{U}\mathbf{U}^T$ the risk (6) becomes

$$R_\epsilon(\sigma\mathbf{U}\mathbf{U}^T) = \mathbb{E}_{\mathbf{x},\mathbf{z}} \left[ \max_{\|\mathbf{e}\|_2 \leq \epsilon} \|\mathbf{H}(\mathbf{x} + \mathbf{e} + \mathbf{z}) - \mathbf{x}\|_2^2 \right]$$

$$= \mathbb{E}_{\mathbf{x},\mathbf{z}} \left[ \max_{\|\mathbf{e}\|_2 \leq \epsilon} \|\sigma\mathbf{U}\mathbf{U}^T(\mathbf{U}\mathbf{c} + \mathbf{e} + \mathbf{z}) - \mathbf{U}\mathbf{c}\|_2^2 \right]$$

$$= \mathbb{E}_{\mathbf{c},\mathbf{z}} \left[ \max_{\|\mathbf{e}_\|\|_2 \leq \epsilon} \|(\sigma - 1)\mathbf{c} + \sigma\mathbf{e}_\| + \sigma\mathbf{z}_\|\|_2^2 \right].$$

With Lemma A.1,

$$R_\epsilon(\sigma\mathbf{U}\mathbf{U}^T) = \min_{\mathbf{H}} \mathbb{E}_{\mathbf{c},\mathbf{z}_\|} \left[ \min_{\lambda : \lambda \geq \sigma^2} \lambda\epsilon^2 + ((\sigma - 1)\mathbf{c} + \sigma\mathbf{z}_\|)^T \left( \mathbf{I} - \frac{1}{\lambda}\sigma^2\mathbf{I} \right)^{-1} ((\sigma - 1)\mathbf{c} + \sigma\mathbf{z}_\|) \right]$$

$$= \min_{\mathbf{H}} \mathbb{E}_{\mathbf{c},\mathbf{z}} \left[ \min_{\lambda : \lambda \geq \sigma^2} \lambda\epsilon^2 + \sum_{i=1}^d \frac{((\sigma - 1)c_i + \sigma z_i)^2}{1 - \frac{\sigma^2}{\lambda}} \right]$$

$$= \min_{\mathbf{H}} \mathbb{E}_{\mathbf{g}} \left[ \min_{\lambda : \lambda \geq \sigma^2} \lambda\epsilon^2 + \sum_{i=1}^d g_i^2 \frac{(\sigma - 1)^2 \frac{\sigma_c^2}{d} + \sigma^2 \frac{\sigma_z^2}{n}}{1 - \frac{\sigma^2}{\lambda}} \right].$$

Using the optimal $\lambda^*$, by equation 10, we get

$$R_\epsilon(\sigma \mathbf{U}\mathbf{U}^T) = \mathbb{E}\left[\left(\epsilon\sigma + \|\mathbf{g}\|_2 \frac{1}{\sqrt{d}}\sqrt{(\sigma-1)^2\sigma_c^2 + \sigma^2 \frac{d}{n}\sigma_z^2}\right)^2\right]$$

$$= \mathbb{E}\left[(\epsilon\sigma)^2 + 2\epsilon\sigma\|\mathbf{g}\|_2 \frac{1}{\sqrt{d}}\sqrt{(\sigma-1)^2\sigma_c^2 + \sigma^2 \frac{d}{n}\sigma_z^2} + \|\mathbf{g}\|_2^2 \frac{1}{d}\left((\sigma-1)^2\sigma_c^2 + \sigma^2 \frac{d}{n}\sigma_z^2\right)\right]$$

$$\overset{\text{i}}{\leq} (\epsilon\sigma)^2 + 2\epsilon\sigma\sqrt{(\sigma-1)^2\sigma_c^2 + \sigma^2 \frac{d}{n}\sigma_z^2} + \left((\sigma-1)^2\sigma_c^2 + \sigma^2 \frac{d}{n}\sigma_z^2\right)$$

$$\overset{\text{ii}}{=} \left(\epsilon\sigma + \sqrt{(\sigma-1)^2\sigma_c^2 + \sigma^2 \frac{d}{n}\sigma_z^2}\right)^2,$$

where equation (i) follows by using Jensen's inequality once again (specifically, using $(\mathbb{E}[\|\mathbf{g}\|_2])^2 \leq \mathbb{E}[\|\mathbf{g}\|_2^2] = d$). Noting that (ii) is equal to the lower bound of the risk for any symmetric $\mathbf{H}$ in equation (11) shows that $\mathbf{H} = \sigma \mathbf{U}\mathbf{U}^T$ with the optimal parameter $\sigma^*$ derived above is optimal.

## A.3   Proof of Lemma A.1

The optimization problem

$$\max_{\|\mathbf{e}\|_2 \leq \epsilon} \|\mathbf{z} - \mathbf{H}\mathbf{e}\|_2^2 \tag{13}$$

can be written as

$$\min_{\mathbf{e}} -\|\mathbf{z} - \mathbf{H}\mathbf{e}\|_2^2 \text{ subject to } \epsilon^2 - \mathbf{e}^T\mathbf{e} \geq 0.$$

The corresponding Lagrangian is, for $\lambda \geq 0$

$$L(\mathbf{e}, \lambda) = -(\mathbf{z} - \mathbf{H}\mathbf{e})^T(\mathbf{z} - \mathbf{H}\mathbf{e}) - \lambda(\epsilon^2 - \mathbf{e}^T\mathbf{e})$$

$$= -\mathbf{z}^T\mathbf{z} + 2\mathbf{z}^T\mathbf{H}\mathbf{e} - \mathbf{e}^T\mathbf{H}^T\mathbf{H}\mathbf{e} - \lambda\epsilon^2 + \lambda\mathbf{e}^T\mathbf{e}$$

$$= \mathbf{e}^T(\lambda\mathbf{I} - \mathbf{H}^T\mathbf{H})\mathbf{e} + 2\mathbf{z}^T\mathbf{H}\mathbf{e} - \lambda\epsilon^2 - \mathbf{z}^T\mathbf{z}$$

The Lagrange Dual Function is

$$q(\lambda) = \inf_{\mathbf{e}} L(\mathbf{e}, \lambda).$$

Using that $\min_{\mathbf{e}} 2\mathbf{c}^T\mathbf{e} + \mathbf{e}^T\mathbf{B}\mathbf{e} = -\mathbf{c}^T\mathbf{B}^{-1}\mathbf{c}$, we get, provided that $\lambda \geq \lambda_{\max}(\mathbf{H}^T\mathbf{H})$,

$$q(\lambda) = -\mathbf{z}^T\mathbf{H}^T(\lambda\mathbf{I} - \mathbf{H}^T\mathbf{H})^{-1}\mathbf{H}^T\mathbf{z} - \lambda\epsilon^2 - \mathbf{z}^T\mathbf{z}$$

$$= -\mathbf{z}^T\left(\mathbf{I} + \mathbf{H}(\lambda\mathbf{I} - \mathbf{H}^T\mathbf{H})^{-1}\mathbf{H}\right)\mathbf{z} - \lambda\epsilon^2$$

$$= -\mathbf{z}^T\left(\mathbf{I} - \frac{1}{\lambda}\mathbf{H}^T\mathbf{H}\right)^{-1}\mathbf{z} - \lambda\epsilon^2$$

where the last equality follows from the Woodburry Identity. Thus, the dual problem is

$$\max_{\lambda \geq 0} q(\lambda) = \min_{\lambda \geq 0} -q(\lambda)$$

$$= \min_{\lambda \geq \lambda_{\max}(\mathbf{H}^T\mathbf{H})} \lambda\epsilon^2 + \mathbf{z}^T\left(\mathbf{I} - \frac{1}{\lambda}\mathbf{H}^T\mathbf{H}\right)^{-1}\mathbf{z}.$$

Even though the primal problem (13) is non-convex, strong duality holds, since the primal program is a quadratic program that is strictly feasible, see Boyd and Vandenberghe (2004, Appendix B.1). The primal problem is strictly feasible if there exists a vector $\mathbf{e}$ such that $\epsilon^2 - \mathbf{e}^T\mathbf{e} > 0$. This is trivially satisfied as long as $\epsilon > 0$.

# B  Additional proofs for denoising

We state two more proofs on optimal worst-case denoisers and jittering.

## B.1  Proof of Corollary 3.2

In the main text it was stated that linear estimators $f(\mathbf{y}) = \mathbf{H}\mathbf{y}$ that minimize the jittering risk $J_{\sigma_w}$ with noise level chosen as a function of the desired noise level $\epsilon$ as

$$\sigma_w(\epsilon) = \sqrt{\frac{\epsilon^2 \sigma_z^2 \frac{d}{n} + \sigma_z \sqrt{\frac{d}{n}} \sigma_c \epsilon \sqrt{\sigma_c^2 - \epsilon^2 + \sigma_z^2 \frac{d}{n}}}{d(\sigma_c^2 - \epsilon^2)}}$$

also minimizes the worst-case risk $R_\epsilon$.

The result follows from Theorem 3.1. For that let $f_r$ and $f_j$ be linear estimators minimizing the robust risk $R_\epsilon$ and the jittering risk $J_{\sigma_w}$, respectively. By Theorem 3.1, the two estimators are scaled projections onto the subspace, i.e., $f_r(\mathbf{y}) = \alpha_r \mathbf{U}\mathbf{U}^T$ and $f_j(\mathbf{y}) = \alpha_j \mathbf{U}\mathbf{U}^T$ with

$$\alpha_r = \frac{\sigma_c^2 - \frac{\epsilon \sigma_c \sigma_z \sqrt{\frac{d}{n}}}{\sqrt{\sigma_c^2 - \epsilon^2 + \sigma_z^2 \frac{d}{n}}}}{\sigma_c^2 + \sigma_z^2 \frac{d}{n}} \quad \text{and} \quad \alpha_j = \frac{\sigma_c^2}{\sigma_c^2 + \sigma_z^2 \frac{d}{n} + \sigma_w^2 d}.$$

Setting $\alpha_r = \alpha_j$ and solving for the standard deviation $\sigma_w$ yields the result.

## B.2  Form of perturbations for estimators $\mathbf{H} = \alpha \mathbf{U}\mathbf{U}^T$

We noted in the main body that for estimators $\mathbf{H} = \alpha \mathbf{U}\mathbf{U}^T$ worst-case perturbations can be computed in closed form for fixed $\mathbf{y}$ and $\mathbf{x}$. We calculate:

$$\begin{aligned}
\hat{\mathbf{e}} &= \arg \max_{\|\mathbf{e}\|_2 \leq \epsilon} \|\mathbf{H}(\mathbf{x} + \mathbf{z} + \mathbf{e}) - \mathbf{x}\|_2^2 \\
&= \arg \max_{\|\mathbf{e}\|_2 \leq \epsilon} \|(\mathbf{H} - \mathbf{I})\mathbf{U}\mathbf{c} + \mathbf{H}\mathbf{z} + \mathbf{H}\mathbf{e}\|_2^2 \\
&= \arg \max_{\|\mathbf{e}\|_2 \leq \epsilon} \|(\alpha \mathbf{U}\mathbf{U}^T - \mathbf{I})\mathbf{U}\mathbf{c} + \alpha \mathbf{U}\mathbf{U}^T \mathbf{z} + \alpha \mathbf{U}\mathbf{U}^T \mathbf{e}\|_2^2 \\
&= \arg \max_{\|\mathbf{e}\|_2 \leq \epsilon} \|(\alpha - 1)\mathbf{U}\mathbf{c} + \alpha \mathbf{U}\mathbf{U}^T \mathbf{z} + \alpha \mathbf{U}\mathbf{U}^T \mathbf{e}\|_2^2 \\
&= \mathbf{U} \arg \max_{\|\mathbf{e}'\|_2 \leq \epsilon} \|(\alpha - 1)\mathbf{c} + \alpha \mathbf{U}^T \mathbf{z} + \alpha \mathbf{e}'\|_2^2 \\
&= \mathbf{U}\epsilon \frac{(1 - \alpha)\mathbf{c} + \alpha \mathbf{U}^T \mathbf{z}}{\|(1 - \alpha)\mathbf{c} + \alpha \mathbf{U}^T \mathbf{z}\|_2}.
\end{aligned}$$

Thus the perturbation points into the direction of the signal plus noise lying in the signal subspace.

# C  Theory for general linear inverse problems

In the following, we consider linear inverse problems $\mathbf{y} = \mathbf{A}\mathbf{x} + \mathbf{z}$, with a linear forward operator $\mathbf{A} \in \mathbb{R}^{m \times n}$ and noise $\mathbf{z} \sim \mathcal{N}(0, \sigma_z^2/m\mathbf{I})$. For denoising ($\mathbf{A} = \mathbf{I}$) we stated an explicit analytical characterization of the optimal worst-case estimator and presented a proof in the appendix. The proof, however, does not generalize in a straightforward manner to more general inverse problems. We conjecture the worst-case optimal linear estimator for large dimensions $d$ and present numerical simulation results. Moreover, we state the proof for the optimal jittering estimator, and demonstrate that it can yield suboptimal worst-case robust estimators in general.

## C.1  Optimal worst-case robust estimator

As formalized by Lemma A.1, the robust-risk (1) of the estimator $f$ can be written as an expectation involving a minimization problem over a single variable (instead of a maximization over an $n$-

dimensional variable, as in the original definition):

$$R_\epsilon(\mathbf{H}) = \mathbb{E}_{\mathbf{v}} \left[ \min_{\lambda \geq \max_i \sigma_i^2} \lambda \epsilon^2 + \mathbf{v}^T (\mathbf{I} - \frac{1}{\lambda} \mathbf{H}\mathbf{H}^T)^{-1} \mathbf{v} \right], \quad \mathbf{v} = (\mathbf{H}\mathbf{A} - \mathbf{I})\mathbf{x} + \mathbf{H}\mathbf{z}. \quad (14)$$

Here, $\sigma_i$ are the singular values of the matrix $\mathbf{H}$. In order to find the optimal robust estimator we wish to solve the optimization problem $\arg\min_{\mathbf{H}} R_\epsilon(\mathbf{H})$. The difficulty in solving this optimization problem is that we can't solve the minimization problem within the expectation (14) in closed form. In order to prove Theorem 3.1 for denoising (i.e., for $\mathbf{A} = \mathbf{I}$) we derived an upper and a matching lower bound of the risks using several unusual applications of Jensen's inequality. The proof does not generalize in a straightforward manner to the more general case where $\mathbf{A} \neq \mathbf{I}$. However, for large $d$, the random variable $\mathbf{v}^T (\mathbf{I} - \frac{1}{\lambda} \mathbf{H}\mathbf{H}^T)^{-1} \mathbf{v}$ concentrates around it's expectation, and thus we conjecture that for large $d$, we can exchange expectation and minimization, which yields:

$$R_\epsilon(\mathbf{H}) = \min_{\lambda \geq \max_i \sigma_i^2} \lambda \epsilon^2 + \mathbb{E}_{\mathbf{v}} \left[ \mathbf{v}^T (\mathbf{I} - \frac{1}{\lambda} \mathbf{H}\mathbf{H}^T)^{-1} \mathbf{v} \right]. \quad (15)$$

Based on equation (15) we derive a characterization of the optimal worst-case estimator. We proceed similar to the denoising case and start with rearranging the terms in expectation:

$$\mathbf{v}^T (\mathbf{I} - \frac{1}{\lambda} \mathbf{H}\mathbf{H}^T)^{-1} \mathbf{v} = ((\mathbf{H}\mathbf{A} - \mathbf{I})\mathbf{x} + \mathbf{H}\mathbf{z})^T (\mathbf{I} - \frac{1}{\lambda} \mathbf{H}\mathbf{H}^T)^{-1} ((\mathbf{H}\mathbf{A} - \mathbf{I})\mathbf{x} + \mathbf{H}\mathbf{z})$$

$$= \mathbf{c}^T ((\mathbf{H}\mathbf{A} - \mathbf{I})\mathbf{U})^T (\mathbf{I} - \frac{1}{\lambda} \mathbf{H}\mathbf{H}^T)^{-1} (\mathbf{H}\mathbf{A} - \mathbf{I})\mathbf{U}\mathbf{c}$$

$$+ 2\mathbf{z}^T \mathbf{H}^T (\mathbf{I} - \frac{1}{\lambda} \mathbf{H}\mathbf{H}^T)^{-1} (\mathbf{H}\mathbf{A} - \mathbf{I})\mathbf{U}\mathbf{c} + \mathbf{z}^T \mathbf{H}^T (\mathbf{I} - \frac{1}{\lambda} \mathbf{H}\mathbf{H}^T)^{-1} \mathbf{H}\mathbf{z}.$$

Now, let $\mathbf{A}\mathbf{U} = \mathbf{W}\boldsymbol{\Lambda}\mathbf{V}^T$ be the singular value decomposition of the matrix $\mathbf{A}\mathbf{U}$ and $\mathbf{H} = \mathbf{U}\mathbf{V}\boldsymbol{\Sigma}\mathbf{W}^T$ the singular value decomposition of $\mathbf{H}$. We then have:

$$\mathbf{H}\mathbf{H}^T = \mathbf{U}\mathbf{V}\boldsymbol{\Sigma}\mathbf{W}^T (\mathbf{U}\mathbf{V}\boldsymbol{\Sigma}\mathbf{W}^T)^T = \mathbf{U}\mathbf{V}\boldsymbol{\Sigma}\mathbf{W}^T \mathbf{W}\boldsymbol{\Sigma}\mathbf{V}^T \mathbf{U}^T = \mathbf{U}\mathbf{V}\boldsymbol{\Sigma}^2 \mathbf{V}^T \mathbf{U}^T$$

$$\mathbf{H}\mathbf{A}\mathbf{U} = \mathbf{U}\mathbf{V}\boldsymbol{\Sigma}\mathbf{W}^T \mathbf{W}\boldsymbol{\Lambda}\mathbf{V}^T = \mathbf{U}\mathbf{V}\boldsymbol{\Sigma}\boldsymbol{\Lambda}\mathbf{V}^T.$$

For the individual parts in the summation it follows:

$$\mathbf{c}^T (\mathbf{H}\mathbf{A}\mathbf{U} - \mathbf{U})^T (\mathbf{I} - \frac{1}{\lambda} \mathbf{H}\mathbf{H}^T)^{-1} (\mathbf{H}\mathbf{A}\mathbf{U} - \mathbf{U})\mathbf{c}$$

$$= \mathbf{c}^T (\mathbf{U}\mathbf{V}\boldsymbol{\Sigma}\boldsymbol{\Lambda}\mathbf{V}^T - \mathbf{U})^T (\mathbf{I} - \frac{1}{\lambda} \mathbf{U}\mathbf{V}\boldsymbol{\Sigma}^2 \mathbf{V}^T \mathbf{U}^T)^{-1} (\mathbf{U}\mathbf{V}\boldsymbol{\Sigma}\boldsymbol{\Lambda}\mathbf{V}^T - \mathbf{U})\mathbf{c}$$

$$= \mathbf{c}^T (\mathbf{U}\mathbf{V}\boldsymbol{\Sigma}\boldsymbol{\Lambda}\mathbf{V}^T - \mathbf{U})^T \mathbf{U}\mathbf{V} (\mathbf{I} - \frac{1}{\lambda} \boldsymbol{\Sigma}^2)^{-1} \mathbf{V}^T \mathbf{U}^T (\mathbf{U}\mathbf{V}\boldsymbol{\Sigma}\boldsymbol{\Lambda}\mathbf{V}^T - \mathbf{U})\mathbf{c}$$

$$= (\mathbf{V}^T \mathbf{c})^T (\boldsymbol{\Sigma}\boldsymbol{\Lambda} - \mathbf{I})(\mathbf{I} - \frac{1}{\lambda} \boldsymbol{\Sigma}^2)^{-1} (\boldsymbol{\Sigma}\boldsymbol{\Lambda} - \mathbf{I})\mathbf{V}^T \mathbf{c} = \sum_{i=1}^{d} \frac{(\sigma_i \lambda_i - 1)^2 c_i}{1 - \frac{\sigma_i^2}{\lambda}}$$

where we define $c_i = \mathbf{v}_i^T \mathbf{c}$ using the $i$-th column vector $\mathbf{v}_i$ of the matrix $\mathbf{V}$. Similarly, we set $z_i = \mathbf{w}_i^T \mathbf{z}$, with $\mathbf{w}_i$ the $i$-th column vector of $\mathbf{W}$, and get for the other parts:

$$2\mathbf{z}^T \mathbf{H}^T (\mathbf{I} - \frac{1}{\lambda} \mathbf{H}\mathbf{H}^T)^{-1} (\mathbf{H}\mathbf{A}\mathbf{U} - \mathbf{U})\mathbf{c} = \sum_{i=1}^{d} 2 \frac{z_i \sigma_i (\sigma_i \lambda_i - 1) c_i}{1 - \frac{\sigma_i^2}{\lambda}}$$

$$\mathbf{z}^T \mathbf{H}^T (\mathbf{I} - \frac{1}{\lambda} \mathbf{H}\mathbf{H}^T)^{-1} \mathbf{H}\mathbf{z} = \sum_{i=1}^{d} \frac{z_i^2 \sigma_i^2}{1 - \frac{\sigma_i^2}{\lambda}}.$$

Hence, for the term in expectation in Eq. (15) we get:

$$\mathbb{E}_{\mathbf{v}} \left[ \mathbf{v}^T (\mathbf{I} - \frac{1}{\lambda} \mathbf{H}\mathbf{H}^T)^{-1} \mathbf{v} \right] = \mathbb{E}_{\mathbf{c},\mathbf{z}} \left[ \sum_{i=1}^{d} \frac{(c_i(\sigma_i \lambda_i - 1) + z_i \sigma_i)^2}{1 - \frac{\sigma_i^2}{\lambda}} \right]$$

$$= \mathbb{E}_{\mathbf{g}} \left[ \sum_{i=1}^{d} \frac{g_i^2 \left( \frac{\sigma_i^2}{d}(\sigma_i \lambda_i - 1)^2 + \frac{\sigma_z^2}{m} \sigma_i^2 \right)}{1 - \frac{\sigma_i^2}{\lambda}} \right],$$

where we note that $c_i(\sigma_i\lambda_i - 1) + z_i\sigma_i$ are iid zero-mean gaussian with variance $\frac{\sigma_c^2}{d}(\lambda_i\sigma_i - 1)^2 + \frac{\sigma_z^2}{m}\sigma_i^2$ and $g_i \sim \mathcal{N}(0,1)$. From this it follows, assuming the robust risk conjecture (15), that the optimal robust estimator minimizes:

$$\min_{\mathbf{H}} R_\epsilon(\mathbf{H}) = \min_{\sigma_i, \lambda \geq \sigma_i^2} f(\lambda, \sigma_1, \ldots, \sigma_n), \tag{16}$$

$$f(\lambda, \sigma_1, \ldots, \sigma_n) = \lambda\epsilon^2 + \sum_{i=1}^{d} \frac{\frac{\sigma_z^2}{m}\sigma_i^2 + \frac{\sigma_c^2}{d}(\sigma_i\lambda_i - 1)^2}{1 - \frac{\sigma_i^2}{\lambda}}. \tag{17}$$

We first calculate the unconstrained minimizer of the function $f$ and get:

$$\frac{\partial f}{\partial \sigma_i} = 2\lambda \frac{\lambda\lambda_i^2\sigma_i\frac{\sigma_c^2}{d} - \lambda_i(\lambda + \sigma_i^2)\frac{\sigma_c^2}{d} + \sigma_i(\frac{\sigma_c^2}{d} + \lambda\frac{\sigma_z^2}{m})}{(\lambda - \sigma_i^2)^2}.$$

For $\lambda_i = 0$ we obtain $\frac{\partial f}{\partial \sigma_i} = 0 \Rightarrow \sigma_i = 0$ and for $\lambda_i \neq 0$:

$$\frac{\partial f}{\partial \sigma_i}\Big|_{\sigma_i = \sigma_{i,\pm}^*} = 0 \Rightarrow \sigma_{i,\pm}^* = \frac{1 + \lambda_i^2\lambda}{2\lambda_i} + \frac{d}{m}\frac{\lambda}{2\lambda_i}\frac{\sigma_z^2}{\sigma_c^2} \pm \sqrt{\left(\frac{1 + \lambda_i^2\lambda}{2\lambda_i} + \frac{d}{m}\frac{\lambda}{2\lambda_i}\frac{\sigma_z^2}{\sigma_c^2}\right)^2 - \lambda}.$$

The solutions $\sigma_{i,\pm}^*$ obtained are for suitable $a(\lambda)$ of the form:

$$\sigma_{i,\pm} = a(\lambda) \pm \sqrt{a(\lambda)^2 - \lambda}.$$

On further examining the constraint $\sigma_i^2 \leq \lambda$ in the problem (16) note that for real numbers $a > b$ it holds $\sqrt{a^2 - b^2} \geq a - b$. Hence, it follows:

$$\sigma_{i,\pm} - \sqrt{\lambda} = a(\lambda) - \sqrt{\lambda} \pm \sqrt{a(\lambda)^2 - \lambda} \gtrless 0.$$

Therefore, we can rule out $\sigma_{i,+}$ and set $\sigma_i^* = \sigma_{i,-}^*$. Finally, inserting $\sigma_i^*$ for $1 \leq i \leq d$ into the robust risk (17) yields:

$$\min_{\mathbf{H}} R_\epsilon(\mathbf{H}) = \min_{\lambda \geq 0} \lambda\epsilon^2 + \sum_{i=1}^{d} \frac{1 - \lambda\lambda_i^2}{2}\frac{\sigma_c^2}{d} - \frac{\lambda}{2}\frac{\sigma_z^2}{m} + \sqrt{\left(\frac{1 + \lambda\lambda_i^2}{2}\frac{\sigma_c^2}{d} + \frac{\lambda}{2}\frac{\sigma_z^2}{m}\right)^2 - \lambda\lambda_i^2\frac{\sigma_c^4}{d^2}}. \tag{18}$$

The optimization problem involved is convex and box-constrained and can thus be solved numerically. Besides the argument above, we confirmed our conjecture with numerical simulations.

## C.2 Optimal jittering estimator

We now derive the optimal jittering estimator, i.e., the estimator $f(\mathbf{y}) = \mathbf{Hy}$ that minimizes the jittering risk

$$J_{\sigma_w}(f) = \mathbb{E}_{(\mathbf{x},\mathbf{z},\mathbf{w})}\left[\|\mathbf{H}(\mathbf{Ax} + \mathbf{w} + \mathbf{z}) - \mathbf{x}\|_2^2\right],$$

where as before the signal $\mathbf{x}$ is assumed to lie within a subspace, i.e., $\mathbf{x} = \mathbf{Uc}$. We first calculate the expectation by using that $\mathbf{c} \sim \mathcal{N}(0, \sigma_c^2/d\mathbf{I})$, $\mathbf{z} \sim \mathcal{N}(0, \sigma_z^2/m\mathbf{I})$ and $\mathbf{w} \sim \mathcal{N}(0, \sigma_w^2\mathbf{I})$ are Gaussian distributed. The jittering risk is

$$J_{\sigma_w}(f) = \mathbb{E}_{(\mathbf{x},\mathbf{z},\mathbf{w})}\left[\|\mathbf{H}(\mathbf{Ax} + \mathbf{z} + \mathbf{w}) - \mathbf{x}\|_2^2\right]$$

$$= \mathbb{E}_{\mathbf{x}}\left[\|(\mathbf{HA} - \mathbf{I})\mathbf{Uc}\|_2^2\right] + \mathbb{E}_{\mathbf{z}}\left[\|\mathbf{Hz}\|_2^2\right] + \mathbb{E}_{\mathbf{w}}\left[\|\mathbf{Hw}\|_2^2\right]$$

$$= \mathrm{tr}\left((\mathbf{HA} - \mathbf{I})\mathbf{UU}^T(\mathbf{HA} - \mathbf{I})^T\right)\frac{\sigma_c^2}{d} + \mathrm{tr}\left(\mathbf{HH}^T\right)\frac{\sigma_z^2}{m} + \mathrm{tr}\left(\mathbf{HH}^T\right)\sigma_w^2.$$

Hence, the optimal jittering estimator minimizes

$$J_{\sigma_w}(f) = \mathrm{tr}(\mathbf{HXH}^T) - 2\,\mathrm{tr}(\mathbf{HY}) + \sigma_c^2,$$

where $\mathbf{X} = \frac{\sigma_c^2}{d}\mathbf{AU}(\mathbf{AU})^T + \left(\sigma_w^2 + \sigma_z^2/m\right)\mathbf{I}$ and $\mathbf{Y} = \frac{\sigma_c^2}{d}\mathbf{AUU}^T$. Using matrix calculus we calculate the optimal estimator as

$$\nabla_{\mathbf{H}} J_{\sigma_w}(f)\Big|_{\mathbf{H}=\mathbf{H}^*} = 2\mathbf{H}^*\mathbf{X} - 2\mathbf{Y}^T = 0,$$

which yields $\mathbf{H}^* = \mathbf{Y}^T\mathbf{X}^{-1}$. Now, let $\mathbf{AU} = \mathbf{W}\boldsymbol{\Lambda}\mathbf{V}^T$ be the singular value decomposition of $\mathbf{AU}$. Then:

$$\mathbf{X} = \frac{\sigma_c^2}{d}\mathbf{W}\boldsymbol{\Lambda}^2\mathbf{W}^T + \left(\sigma_w^2 + \sigma_z^2/m\right)\mathbf{I}$$

$$\mathbf{Y} = \frac{\sigma_c^2}{d}\mathbf{W}\boldsymbol{\Lambda}\mathbf{V}^T\mathbf{U}^T.$$

We get the optimal jittering estimator as:

$$\mathrm{argmin}_{\mathbf{H}}\, J_{\sigma_w}(f) = \mathbf{Y}^T\mathbf{X}^{-1} = \frac{\sigma_c^2}{d}\mathbf{U}\mathbf{V}\boldsymbol{\Lambda}\mathbf{W}^T \left(\frac{\sigma_c^2}{d}\mathbf{W}\boldsymbol{\Lambda}^2\mathbf{W}^T + \left(\sigma_w^2 + \sigma_z^2/m\right)\mathbf{I}\right)^{-1} \quad (19)$$

$$= \frac{\sigma_c^2}{d}\mathbf{U}\mathbf{V}\boldsymbol{\Lambda}\left(\frac{\sigma_c^2}{d}\boldsymbol{\Lambda}^2 + \left(\sigma_w^2 + \sigma_z^2/m\right)\mathbf{I}\right)^{-1}\mathbf{W}^T \quad (20)$$

$$= \mathbf{U}\mathbf{V}\mathrm{diag}\left(\frac{\sigma_c^2\lambda_i}{\sigma_c^2\lambda_i^2 + d\sigma_w^2 + \sigma_z^2\frac{d}{m}}\right)\mathbf{W}^T. \quad (21)$$

### C.3 Numerical simulations supporting Conjecture 3.3

In the following we present numerical simulations for general linear inverse problems in the subspace model to support the Conjecture 3.3 further. The results show that the (empirical) optimal robust risk, obtained via adversarial training, is the same as the robust risk of the conjectured optimal estimator. Moreover, the results show that jittering can yield suboptimal robust estimators in some cases.

We consider linear inverse problems $\mathbf{y} = \mathbf{Ax} + \mathbf{z}$ with the signal $\mathbf{x}$ lying in a subspace $\mathbf{x} = \mathbf{Uc}$ with $\mathbf{c} \sim \mathcal{N}(0, \sigma_c^2/d\mathbf{I})$ and Gaussian noise $\mathbf{z} \sim \mathcal{N}(0, \sigma_z^2/n\mathbf{I})$. For the simulations, we choose $\sigma_c/\sqrt{d} = 1$ and $\sigma_z/\sqrt{n} = 0.1$ with dimensions $d = 50$ and $n = 100$. Moreover, we consider two diagonal forward operators $\mathbf{A} = \mathrm{diag}(\lambda_i)$: an operator with linear decaying singular values ($\lambda_i = \frac{i}{n}$) and one with geometrically decaying singular values $\lambda_i = 0.7^i$.

**Optimal worst-case estimator.** We first estimate the optimal robust risks by performing adversarial training and compare it with the robust risk of the conjectured optimal estimator. Adversarial training is performed as described in Section 4, where we generate data $\{(\mathbf{y}_i, \mathbf{x}_i)\}$ using the respective forward model in the subspace. The estimator $f(\mathbf{y}) = \mathbf{Hy}$ can be viewed as a neural network with one layer and without bias. Figure 5 shows that the robust risk of the conjectured estimator is essentially the same as the (empirical) optimal robust risk.

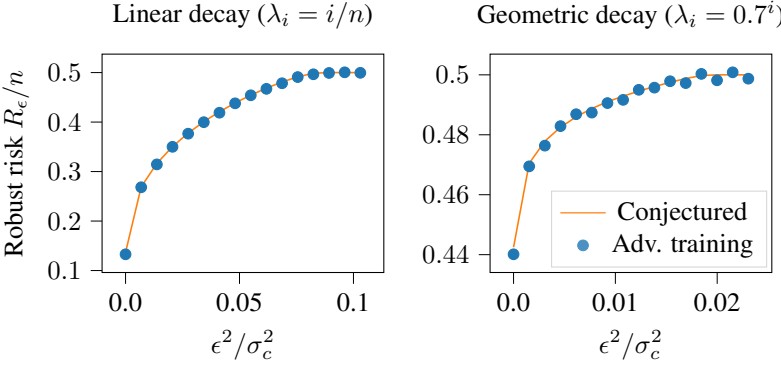

Figure 5: Robust risks of adversarial training (dots) and the conjectured optimal estimator (lines) for two forward operators within the subspace model. It can be seen that the conjectured estimators yield essentially the same robust risks as those of adversarial training.

**Suboptimality of jittering.** We further investigate whether optimal robust estimators can be obtained via jittering. To that end, we calculate the minimal robust risk attainable via jittering

$$\min_{\sigma_w} R_\epsilon(\operatorname{argmin}_f J_{\sigma_w}(f)),$$

where $f(\mathbf{y}) = \mathbf{H}\mathbf{y}$ is a linear reconstruction operator as before. We make use of the analytic expression of the jittering estimator in Eq. (21) and calculate the attainable robust risk by minimizing the robust risk of jittering with respect to the jittering noise level. The results are compared to the (conjectured) optimal robust risks via Eq. (18) and the robust risk of the standard estimator. Figure 6 shows the results of the calculations for the forward operator with linear and geometrically decaying singular values described above. It can be seen that the standard estimator is noticeable less robust compared to denoising setups. Moreover, a gap can be observed between the robust risk obtained via jittering and the optimal robust risks.

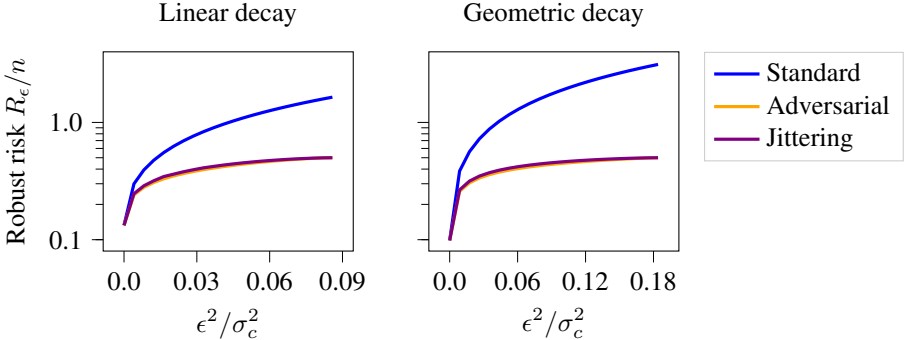

Figure 6: Pixel-wise robust risk of the optimal worst-case estimator, the jittering estimator with optimal jittering noise level, and the standard estimators. A small gap can be observed, showing that (isotrope) jittering does not yield the optimal linear worst-case estimator in general. Moreover, the standard estimator is more sensitive to adversarial perturbations as in the denoising setup.

## D  Details on the experimental results and further experimental results

In this section, we present details on the experimental results in Section 4, and present further experimental results on U-nets trained with robustness-enhancing methods for image denoising, deconvolution, and compressive sensing. The setup and methods considered are as described in the main body.

### D.1  Optimal robust denoiser for large perturbation levels

In the main body, we presented empirical results for perturbation levels in the range $\epsilon^2/\mathbb{E}\left[\|\mathbf{x}\|_2^2\right] \in [0, 0.3]$, since this is the practically most relevant regime. Figure 7 shows the risk of linear estimators and U-nets trained adversarially for perturbation levels in the range $\epsilon^2/\mathbb{E}\left[\|\mathbf{x}\|_2^2\right] \in [0, 1.5]$. From those plots, we see that the transition at $\epsilon^2/\mathbb{E}\left[\|\mathbf{x}\|_2^2\right] = 1$ predicted by Theorem 3.1 for the estimator to map to zero occurs for the subspace model (as predicted by the theory) as well as for the U-net for image denoising.

### D.2  Convolution kernel for deconvolution experiments

In addition to experiments on denoising image data, we consider a deconvolution setup $\mathbf{y} = \mathbf{k} \star \mathbf{x} + \mathbf{z}$ in this work. The kernel $\mathbf{k}$ is Gaussian, applied channel-wise and visualized in Figure 8.

### D.3  Quantifying the smoothing effect of Jittering

In the main text we presented visual example reconstructions and observed that neural networks trained via Jittering yield smoother reconstructions compared to adversarial and standard training.

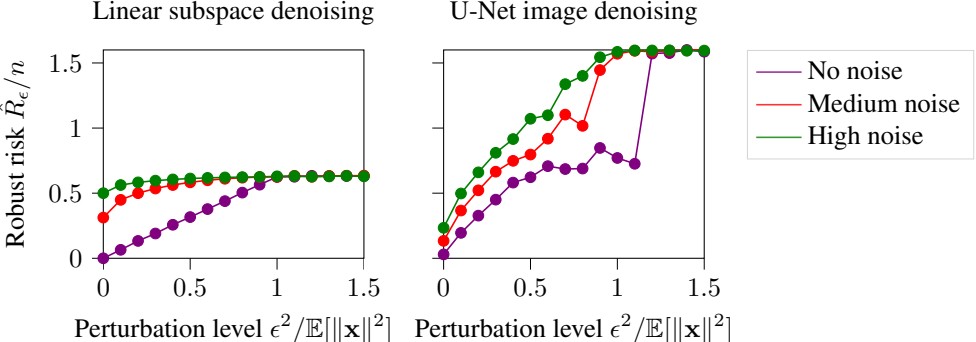

Figure 7: The empirical robust risk for models trained with adversarial training for noise levels $\sigma_z = 0$ (no noise), $\sigma_z/\sqrt{n} = 0.5$ (medium noise) and $\sigma_z/\sqrt{n} = 1.5$ (high noise). The transition predicted by Theorem 3.1 at $\epsilon^2 \approx \mathbb{E}\left[\|\mathbf{x}\|_2^2\right]$ for the estimator to map to zero occurs for the subspace and image denoising settings.

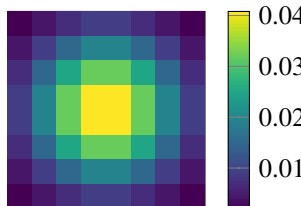

Figure 8: **Gaussian kernel for the deconvolution experiment.** The $8 \times 8$-pixel kernel is calculated as discretization of the $2d$ Gaussian density with standard deviation 2.

We further quantify this observation by calculating the total variation norm of reconstructed images for the deconvolution task. The results are shown Figure 9 and confirm the observation that networks trained via Jittering yield smoother images.

### D.4 Hyperparameter selection

In the experiments we treat the Jittering noise level $\sigma_w$ as a hyperparameter, which we optimize over a validation dataset to obtain robust estimators at the desired perturbation levels. The hyperparameter search is performed by choosing a grid of jittering noise levels for each task. For each noise level neural networks (U-Nets) are trained via Jittering, and subsequently evaluated on the considered perturbation levels. Figures 10, 11 and 12 show the robust risks of jittering for the considered tasks, as well as the derived jittering choice rule. The smooth curves on the left panels represent the robust risk at a particular robustness level and are obtained by applying uniform filters on the evaluation results. It can be seen that for image denoising the empirical jittering choice is close to the prediction from theory.

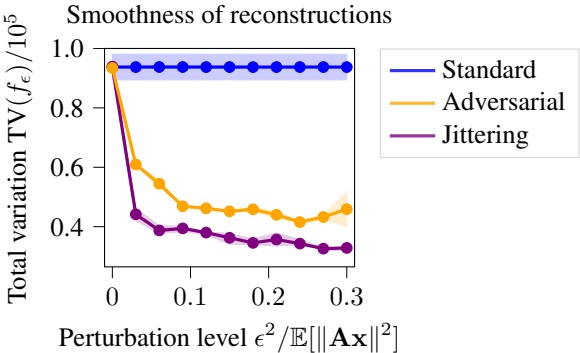

Figure 9: **Jittering yields smoother reconstructions for deconvolution.** The plot shows the total variation norm of images, which are reconstructions using U-nets $f_\epsilon$ trained via jittering, adversarially and via standard training. The TV-norm is calculated per reconstructed image $f_\epsilon(\mathbf{y}_i)$ and averaged over the dataset, i.e. $\text{TV}(f_\epsilon) = \frac{1}{n} \sum_{i=1}^{n} \|f_\epsilon(\mathbf{y}_i)\|_{\text{TV}}$, with $n = 2000$ measurements $\mathbf{y}_i$.

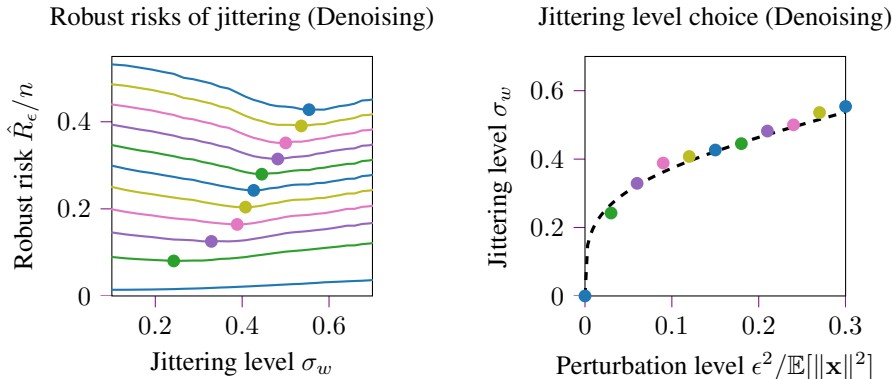

Figure 10: **Jittering hyperparameter results for image denoising.** Neural networks (U-nets) are trained via jittering on a grid of jittering noise levels $\sigma_w$ (with fixed noise level $\sigma_z/\sqrt{n} = 0.25$). The models are subsequently evaluated on the robust risk $R_\epsilon$, which is calculated for each of the perturbation levels $\epsilon$. Each line in the left panel corresponds to the robust risk at one perturbation level. The derived choice rule is displayed in the right panel. It can be seen that for denoising the empirical jittering choice matches the prediction (dashed line) from theory well (Corollary 3.2, with $n = 128 \cdot 128 \cdot 3 = 49152$ and subspace dimension $d = 32000$).

### D.5 Computational complexity

We measured the GPU time until convergence and memory utilization of the robustness-enhancing schemes on the task of Gaussian denoising of colorized images. Figure 13 shows the training error of adversarial training, training via jittering and standard training as a function of the number of epochs. It shows networks trained at two perturbation levels for adversarial training and jittering (parameter choice taken from Figure 10). We find that all methods require a similar number of epochs for convergence (roughly 600 epochs). Table 1 presents the measured GPU time until convergence and average memory consumption. It can be seen that adversarial training is by a factor of the projected gradient ascent steps (3 in this plot) more expensive than jittering. Moreover, training via jittering has similar computational cost as standard training in terms of GPU time. All three methods require a similar amount of GPU memory to train.

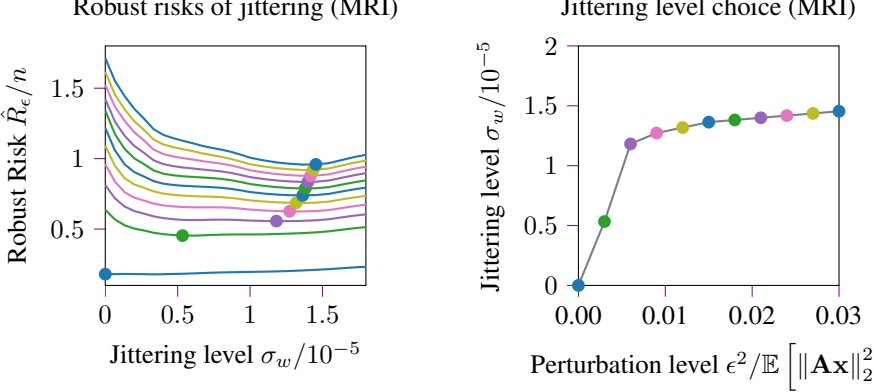

Figure 11: **Jittering hyperparameter search for compressive sensing.** Neural networks (U-nets) are trained via jittering on a grid of noise levels $\sigma_w$. The models are subsequently evaluated on the robust risk $R_\epsilon$, which is calculated for each of the perturbation levels $\epsilon$. Each line in the left panel corresponds to the robust risk at one perturbation level. The derived choice rule is displayed in the right panel.

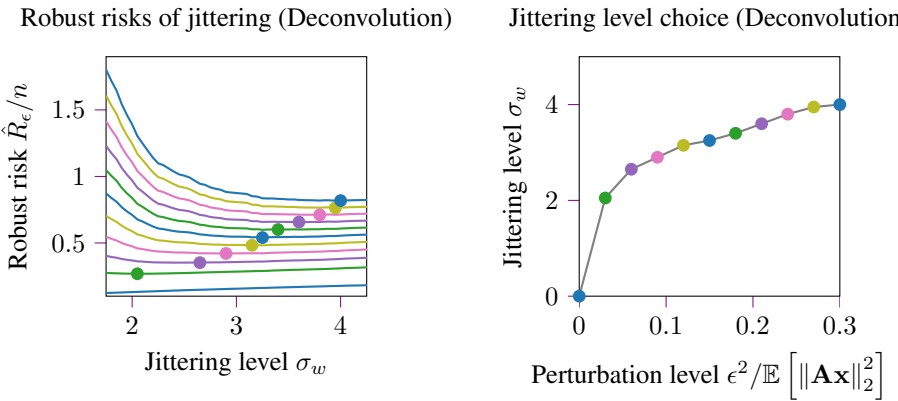

Figure 12: **Jittering hyperparameter results for image deconvolution.** Neural networks (U-nets) are trained via jittering on a grid of jittering noise levels $\sigma_w$ (with fixed noise level $\sigma_z/\sqrt{n} = 0.25$). The models are subsequently evaluated on the robust risk $R_\epsilon$, which is calculated for each of the perturbation levels $\epsilon$. Each line in the left panel corresponds to the robust risk at one perturbation level. The derived choice rule is displayed in the right panel.

### D.6 Required compute

The experimental results presented in this paper were computed using on-premise infrastructure equipped with Nvidia RTX A6000 GPUs. In total, 18 GPU days were spent for the presented colorized image denoising, 20 GPU days for colorized image deconvolution and 15 GPU days for the compressive sensing experiments.

## E Discussion of the related work on randomized smoothing

Randomized smoothing is a very successful technique for obtaining robust classifiers (Cohen et al., 2019; Carlini et al., 2023). Randomized smoothing constructs a smoothed classifier based on a base classifier by averaging the base classifier's outputs under Gaussian noise perturbation. The smoothed classifier allows for certified radii in which it is provably robust, without making any restrictions on the base classifier. However, Salman et al. (2020) demonstrated that it can give loose bounds, since the base classifier is not trained to be robust to Gaussian noise. For that reason Salman et al.

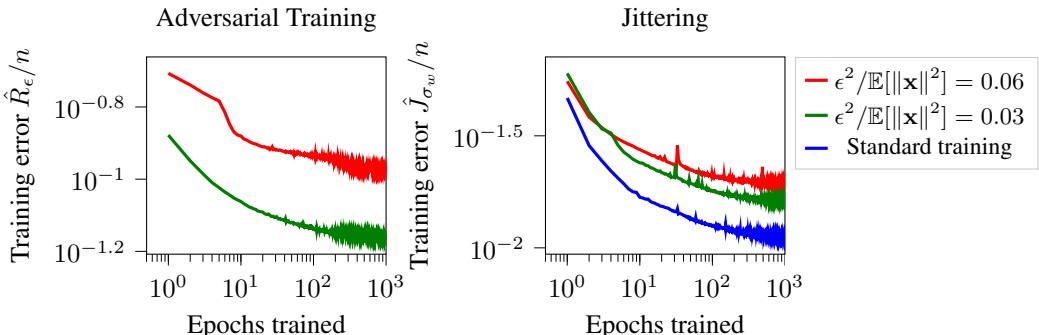

Figure 13: Training metrics of U-nets for adversarial training, jittering and standard training. The plot shows that the convergence rates are very similar for the considered setup. The curves show the training of networks for perturbation levels $\epsilon^2/\mathbb{E}[\|\mathbf{x}\|^2] = 0.03$ (red) and $0.06$ (green). Training the methods takes roughly 600 epochs for convergence.

| Method | Total GPU hours | Memory |
|---|---|---|
| Adversarial training | 12.2 h | 9971 MiB |
| Training via jittering | 3.0 h | 9523 MiB |
| Standard training | 3.0 h | 9523 MiB |

Table 1: GPU time until convergence and memory utilization of adversarial training, jittering and standard training for the task of denoising colorized images. The required number of epochs is estimated from the data visualized in Figure 13. GPU time and memory consumption are measured on a Nvidia RTX A6000 GPU.

(2020) propose denoised smoothing, which considers a composition of the base classifier with a denoising method. At first sight, randomized smoothing might sound similar to the Jittering approach investigated here. However, as we argue below, randomized smoothing is conceptually very different from Jittering.

Given a classifier $f\colon \mathbb{R}^d \to \{1,\dots,K\}$ randomized smoothing constructs a smoothed classifier $g$ from the classifier $f$ as:

$$g(\mathbf{x}) = \arg \min_{k\in\{1,\dots,K\}} \mathbb{E}_{\mathbf{e}\sim\mathcal{N}(0,\sigma^2\mathbf{I})}\left[\mathbb{1}\{f(\mathbf{x}+\mathbf{e})\neq k\}\right],$$

where the parameter $\sigma^2$ controls the robustness-accuracy tradeoff.

For an inverse problem, where we aim to reconstruct a signal $\mathbf{x} \in \mathbb{R}^n$ from a measurement $\mathbf{y} \in \mathbb{R}^m$ using a given reconstruction method $f$, replacing the $0/1$ by the $\ell_2$ loss yields:

$$g(\mathbf{y}) = \arg \min_{\mathbf{x}} \mathbb{E}_{\mathbf{e}\sim\mathcal{N}(0,\sigma^2\mathbf{I})}\left[\|f(\mathbf{y}+\mathbf{e}) - \mathbf{x}\|_2^2\right]$$
$$= \mathbb{E}_{\mathbf{e}\sim\mathcal{N}(0,\sigma^2\mathbf{I})}\left[f(\mathbf{y}+\mathbf{e})\right].$$

For a linear estimator $f(\mathbf{y}) = \mathbf{H}\mathbf{y}$ we see that $g(\mathbf{y}) = f(\mathbf{y})$, so for the linear setting considered in the theory part of this paper randomized smoothing would not change the original estimator. If one considers $f\colon \mathbb{R}^n \to [0,1]^n$ the smoothed estimator $g(\mathbf{y})$ differs from $f(\mathbf{y})$ and robustness gains can be expected, which follows from Salman et al. (2019), Lemma 1. In summary, randomized smoothing is very different to jittering in that it constructs a surrogate smoothed model $g(\mathbf{y})$ based on a given fixed estimator $f(\mathbf{y})$, whereas jittering is a training technique.

## F   Regularizing beyond jittering for enhancing robustness

Training neural networks via jittering, with noise levels chosen for larger perturbations, yields smoother reconstructions compared to adversarial training (see results in Section 4). In this section,

we investigate two related regularization methods, $\ell_2$-regularization and Jacobian regularization, discuss the connection to regularization with jittering, and present experimental results for denoising grayscale images.

## F.1 $\ell_2$- and Jacobian regularization in the subspace model

In the subspace model we established that the optimal jittering estimator is also worst-case optimal, when using a suitable choice of noise level $\sigma_w(\epsilon)$. It turns out, that jittering can further be approximated with an explicit regularizer, Jacobian regularization. For the linear setup considered here, this approximation becomes exact and therefore Jacobian regularization also enables training a worst-case robust estimator. Specifically, using the linear approximation of the function $f$ around the point $\mathbf{y}$, we get

$$\mathbb{E}_{\mathbf{w}}\left[\|f(\mathbf{y}+\mathbf{w})-\mathbf{x}\|_2^2\right] \approx \mathbb{E}_{\mathbf{w}}\left[\|f(\mathbf{y})+\mathbf{J}_{\mathbf{y}}\mathbf{w}-\mathbf{x}\|_2^2\right] = \|f(\mathbf{y})-\mathbf{x}\|_2^2 + \sigma_w^2\|\mathbf{J}_{\mathbf{y}}\|_F^2. \tag{22}$$

Here, $\mathbf{J}_{\mathbf{y}}$ is the Jacobian of the function $f$ at $\mathbf{y}$. The approximation is good for small values of the noise variance $\sigma_w^2$, and is exact for the linear estimator $f(\mathbf{y}) = \mathbf{H}\mathbf{y}$ we consider in this section. The approximate relation (22) motivates the Jacobian regularized risk, defined as

$$\mathrm{Jac}_\lambda(f) = \mathbb{E}_{(\mathbf{x},\mathbf{y})}\left[\|f(\mathbf{y})-\mathbf{x}\|_2^2 + \lambda\|\mathbf{J}_{\mathbf{y}}\|_F^2\right] \tag{23}$$

The connection between jittering and Jacobian regularization is well known in the literature and discussed by Reed et al. (1995). Recall that for the linear estimator considered in this section the approximation in equation (22) is exact, and therefore Jacobian regularization is equivalent to jittering. Thus, Jacobian regularization yields a provably robust estimator, if the regularization parameter is chosen as $\lambda = \sigma_w^2(\epsilon)$ according to corollary 3.2.

For the linear case, Jacobian regularization is even equivalent to $\ell_2$-regularization, since the Jacobian of the function $f(\mathbf{y}) = \mathbf{H}\mathbf{y}$ is $\mathbf{J}_{\mathbf{y}} = \mathbf{H}$, and thus even $\ell_2$-regularization yields a robust estimator.

## F.2 Experimental results on grayscale image denoising

In the following we present results on Gaussian denoising of grayscale images. While $\ell_2$ regularization is equivalent to jittering in the subspace model, we find that the parameter choice $\lambda = \sigma_w^2(\epsilon)$ does not yield robust neural networks using $\ell_2$ regularization. In contrast, Jacobian regularization turns out to be quite effective for learning neural network denoisers, but is computationally demanding compared to jittering.

### F.2.1 Problem setup

We consider once again the dataset of natural image and convert the images to grayscale. We perform Gaussian denoising, i.e. the problem is to reconstruct the image $\mathbf{x}$ from a measurement $\mathbf{y} = \mathbf{x} + \mathbf{z}$, with $\mathbf{z} \sim \mathcal{N}(0, \sigma_z^2/n\mathbf{I})$ and $\sigma_z/\sqrt{n} = 0.2$.

The estimators are chosen as neural networks (U-nets) with the same architecture as for colorized images. We consider adversarial training, jittering and standard training as baseline and compare against $\ell_2$ and Jacobian regularization:

$\ell_2$ **regularization.** Implemented as weight-decay in PyTorch's SGD optimizer to minimize

$$\hat{W}_\lambda(\boldsymbol{\theta}) = \sum_{i=1}^N \|f_{\boldsymbol{\theta}}(\mathbf{y}_i)-\mathbf{x}_i\|_2^2 + \lambda\|\boldsymbol{\theta}\|_2^2.$$

**Jacobian regularization.** We train networks with the Jacobian regularized empirical risk

$$\widehat{\mathrm{Jac}}_\lambda(\boldsymbol{\theta}) = \sum_{i=1}^N \|f_{\boldsymbol{\theta}}(\mathbf{y}_i)-\mathbf{x}_i\|_2^2 + \lambda\|\mathbf{J}_{\mathbf{y}_i}\|_F^2,$$

where $\mathbf{J}_{\mathbf{y}_i}$ is the Jacobian of the network $f_\theta$ with respect to it's input (not it's parameters) at $\mathbf{y}_i$. This regularization can be viewed as an approximation of the jittering risk, as described in

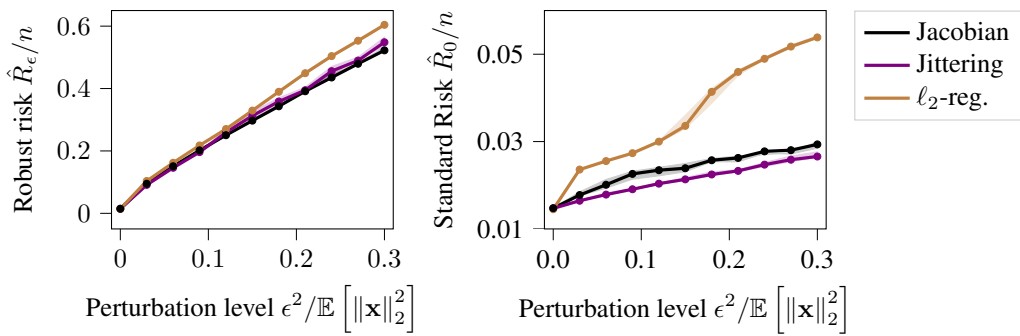

Figure 14: Pixel-wise robust (left) and standard risk (right) of U-nets trained via jittering, $\ell_2$ and Jacobian regularization on the task of denoising grayscale images (at noise level $\sigma_z/\sqrt{n} = 0.2$). The plot shows that $\ell_2$ regularization yields less robust and accurate estimators compared to jittering. In contrast, Jacobian regularization obtains similarly robust estimators, but with slightly weaker performance in standard risk.

subsection F.1.Calculating the full Jacobian $\mathbf{J}_{\mathbf{y}_i}$ with PyTorch requires $n$-many calls of the backward function, which is very expensive, since $n$ is large. To mitigate this cost, we approximate the norm of the Jacobian, $\|\mathbf{J}_{\mathbf{y}_i}\|_F^2$ with $\left\|\mathbf{J}_{\mathbf{y}_i}^T\mathbf{w}\right\|_2^2$, where $\mathbf{w} \sim \mathcal{N}(0, \mathbf{I})$. This approximation of the norm concentrates around the actual squared norm of the Jacobian, and only costs one call of the PyTorch-backward function.

For the experiments we use stochastic gradient descent (SGD) with learning rate $10^{-2}$, momentum $0.9$ and batch size $100$. We evaluate using the empirical pixel-wise robust risk $\hat{R}_\epsilon/n$.

### F.2.2 Results

The experimental results, plotted in Figure 14, show that the networks trained with jittering and Jacobian regularization have similar robust risks compared to the adversarial trained one. Weight-decay or $\ell_2$ regularization yields worse performing estimators than jittering and Jacobian regularization. While for the linear subspace setting, adversarial training, Jacobian and $\ell_2$ regularization are equivalent, for Gaussian denoising they perform differently. Figure 15 shows that Jacobian regularization, unlike jittering, does not yield smoothed images for larger perturbations. However, Jacobian regularization requires approximately $1.4$ as much GPU memory and $4$ times more time per epoch.

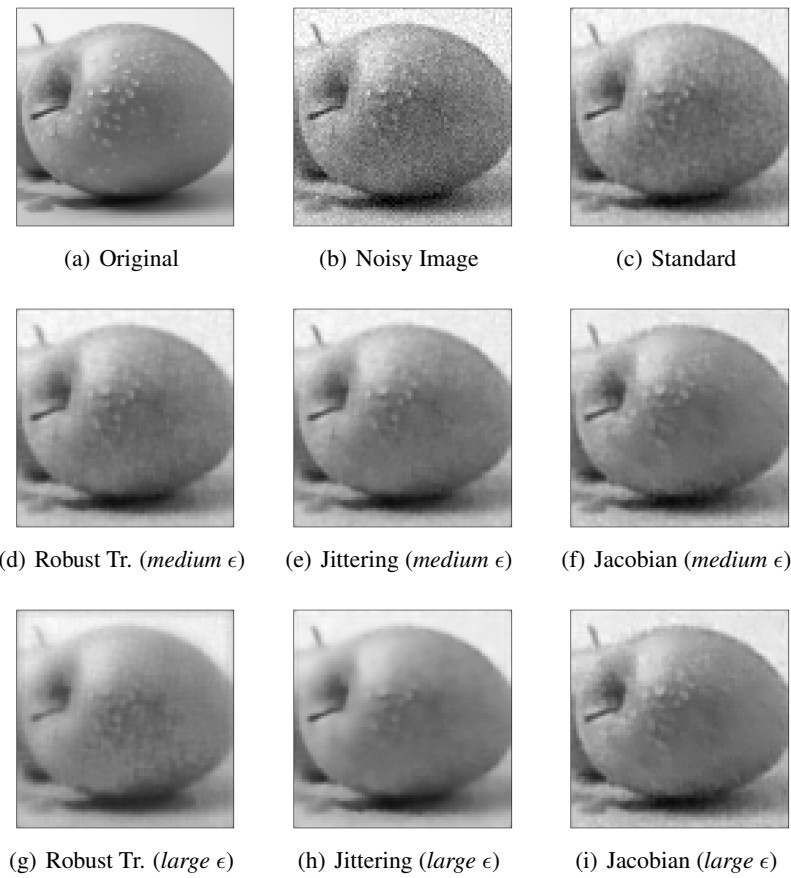

Figure 15: Example reconstructions using U-Nets, trained with different robustness-enhancing schemes at noise level $\sigma_z/\sqrt{n} = 0.2$. The second row depicts the results of using neural networks trained with methods tuned on a *medium* perturbation level of $\epsilon^2/\sigma_c^2 = 0.03$, whereas the third row shows results for a *large* perturbation level $\epsilon^2/\sigma_c^2 = 0.3$. The plot shows that jittering yields smooth reconstructions, whereas adversarial training and Jacobian regularization yield less smooth reconstructions.

