# OpenReview forum: "Learning Provably Robust Estimators for Inverse Problems via Jittering"
_NeurIPS.cc/2023/Conference — NeurIPS 2023 poster_

### Official Review · Reviewer_uuzi · 2023-07-04

**Soundness:** 3 good
**Presentation:** 2 fair
**Contribution:** 3 good
**Rating:** 5
**Confidence:** 3

**Summary:**

This work considers introducing Jittering to the inverse problems for the benefits of robustness to the l2-worst-case. In this work, some analytical results are provided in proving the robust estimation with some assumptions, which I think are reasonable. Then, empirical experiments verify the effectiveness on the robustness improvement at a relatively lower computational cost compared to adversarial training.

**Strengths:**

1. The motivation of this work is quite nice, towards a provable robust estimator.
2. The assumptions are reasonable.
3. Existing results show somewhat promising performance.

**Weaknesses:**

1. Though some analytical results are presented, the actual realization and numerical implementation of the algorithm  are not articulated well I think, e.g., the assumed energy level of signals and noises are normally unknown, so that more details and algorithmic implementation discussions can be presented in a more systematic way, rather than just being mentioned in fractures in Section 4.
2. Several experiments are conduced, but the presented results are limited in each experiment, e.g., cherry pick can be done if only a few images are presented.
3. Some parts of the paper I read a few times, but I still found a bit difficult back and forth to relate all analytical results to the experiments in practical applications.

**Questions:**

1. Some statements lack clarity, e.g., ‘we study jittering, a simple regularization technique that adds noise during training as a robustness-enhancing technique for inverse problems’……as a regularization technique, I would expect a concrete implementation or algorithmic description to make the paper more readable to general readers. (Similar as the bullet 3 in *Weakness);

2. Line 90-94, page 3, [1][2] are mentioned. I quickly checked [1], and it seems like it already implemented the Jittering to the task of inverse problem. It would be nice to provide more details on the difference and relation to [1]. Though [1] has been cited, no further explanations or comparisons are provided later on in this work. As far as I have understood, this work is posed as a step forward upon [1] with some analytical results? This I believe should be clarified.

3. It mentioned the random smoothing method in Section 2 and also the appendix. I understand that RS is significantly different this work. In RS, lots of samplings are required in evaluations. I’m wondering as an expectation over ‘w’ as in (2), there should also be some procedures requiring samplings either for training or evaluations or for sufficing assumptions. Could you please elaborate this part? (Similar as the bullet 3 in *Weakness)

4. In the analytical results of this work, $\sigma_c$, $\sigma_z$ are the essentials. Could you summarize or provide  a general procedure of the relevant implementation procedures or simulation verifications on these variables when they are unknown in the practical settings.  (Similar as the bullet 3 in *Weakness), and the same applies to Section 3.3.

5.  In section 4, besides the bullet 2 in *Weakness, the authors can provide some synthetic experiments to show and explain more detailed results or analysis, as these mentioned variables in the assumptions can be known explicitly and also at least can help the readability.

6. It seems that the hyper-parameter tuning can be time consuming, if too many searches are needed to bring good results.

7. One major claimed benefit is the efficiency, however, only Table 1 is provided roughly, where the complexity analysis is lacking. It should also be nice to mention the computational part involved in finding the hyper parameters. I roughly checked the codes of defining “param” argument and it seems that it just pre-gave a  energy value and heuristically proceeds the setup of other relevant params to run the training. (Similar as the bullet 3 in *Weakness) . For instance, though the compared the adversarial training is more expensive, it doesn't need too many pre-requisite or tuning and can be flexibly applied in various practical tasks.

If this paper can be put in a clearer way connecting the theoretical results and the implementation, and also the empirical experiments can be more practically feasible and convincing in the results, this work would deserve an improved score and I would like to increase my evaluation.

[1] M. Genzel, J. Macdonald, and M. Marz. Solving Inverse Problems With Deep Neural Networks- Robustness Included. IEEE TPAMI 2022
[2] K. V. Gandikota, P. Chandramouli, and M. Moeller. On adversarial robustness of deep image deblurring. In IEEE International Conference on Image Processing, 2022.

**Limitations:**

The authors should verify how convenient and efficient the proposed method can be applied in practical settings and explain some pipelines, in line with the claimed provability of the robust estimator. It seems unclear to me that how promising it could be in generally practical applications, especially considering the claimed analytical aspects.

---

> ### Author Rebuttal · Authors · 2023-08-09
>
> Thanks for the feedback and noting that the motivation is quite nice, the assumptions are reasonable, and the results show promising performance.
>
> Comments on the weaknesses:
> - **Relation of theory and experiment:** The theory gives insights into worst-case robustness and justifies using jittering for obtaining optimal robust estimators. Our theory even predicts the optimal choice of jittering noise level for Gaussian denoising well. In addition, in practice, the optimal jittering level can be determined via a single-parameter hyperparameter search.
> - **Experimental results:** We perform experiments for several inverse problems (denoising, deconvolution and compressive sensing) and evaluate the methods on large datasets. Therefore the results are not cherry picked, as we evaluate on thousands of examples (see 4.1. Problem setup). Perhaps this impression can arise since we show a few example images only. Those images are chosen randomly to visualize reconstructions and to illustrate that jittering yields smoother reconstructions. During the rebuttal, we also quantified the smoothness using the TV norm (see the attached pdf) to be more precise.
>
> Regarding the questions:
> - **Question 1, implementation details on jittering:** Thanks for your feedback on this. It is indeed important to be specific on the implementation details of training with jittering. To fix this, we added the following lines in the paragraph *Training methods* in section 4.1: “Jittering is practically implemented via performing the SGD update rule $\theta \leftarrow \theta - \frac{\eta}{n} \sum_{i=1}^n \nabla_{\theta} \| f_{\theta} (y_i + w_i) - x_i \|^2$. For jittering, the network output is calculated on the noisy input $y_i + w_i$, instead of $y_i$ (standard training) or $y_i + e_i$ (adversarial training). To approximate the expectation we draw independent jittering noise samples $w_i$ in each iteration of SGD.
> - **Question 2, related work on jittering for inverse problems [1,2]:**
> The focus of the paper [1] is to obtain high-quality CT reconstruction. While the paper [1] mentions that it trains with jittering and this improves robustness, there is no systematic study on the effectiveness of jittering, no comparison to robust training, and no theory. In order to clarify the relation to [1,2], we revised the paragraph on jittering for enhancing robustness in inverse problems in our related work section accordingly.
> - **Question 3, regarding sampling and RS**: Randomized smoothing (RS) and jittering both approximate expectations w.r.t. Gaussian random variables. However, since RS is performed during evaluation (fixed network) lots of samples are required to approximate this expectation. Contrary, jittering is performed during training. While every sample is revisited multiple times during multiple epochs of training, it is sufficient to sample new noise at each iteration of SGD. We will further discuss this in the paper.
> - **Question 4 and 5, estimating signal energies in the experiments**: In practical experiments $\sigma_c$ and $\sigma_z$ are not actually required, since the optimal jittering level for robustness can simply be found via hyperparameter search (see also above). If needed (e.g. for testing the theory), they can be calculated as follows:
> *Signal energy*: Given a dataset of $n$ images $x_i$, the signal energy $\sigma_c^2 = \mathbb{E}[\|x\|^2]$ can be estimated via $\frac{1}{n} \sum_{i=1}^n \| x_i \|^2$.
> *Noise level*: If the forward operator $A$ for the inverse problem $y = A x + z$ is known, the noise level can be estimated via $\sigma_z^2 = \mathbb{E}[\|y - A x \|^2]$. If not, a linear reconstruction operator can be trained, and $\sigma_z^2 d/m$ be obtained from its standard risk (see e.g. the formula in 3.2.2. Robustness accuracy trade-off).
> - **Question 6 and 7, cost of hyperparameter search:** The hyperparameter search is computationally inexpensive, since it is only a search over a single scalar variable. Moreover, our experiments show that only relatively few epochs are needed to find the optimal jittering noise levels (e.g. 30 epochs for denoising natural images). The actual training of the networks, however, needs a lot more epochs for convergence (600 for denoising). Thus, the cost for the hyperparameter search is a fraction of the training cost and thus relatively inexpensive.
> Moreover, adversarial training requires $N$ times more GPU hours compared to standard training, where $N$ is the number of iterations for seeking the worst-case examples during training. For the networks depicted in the complexity analysis $N=3$, so adversarial training is three times more expensive (section D.4).
> Due to this scaling, hyperparameter search for jittering is significantly more efficient. Finally, the pre-given energy level can be easily computed from the dataset, as described above. It is used in the code to scale the relative perturbation levels ($\epsilon^2 / \mathbb{E}[\|x\|^2]$ in paper).
>
> We hope that those clarifications and changes connect the theoretical results and the implementation better, and we hope that we clarified that the experiments are practically feasible. Thanks for being open to increasing your score. Please let us know if you have any additional comments and questions.

---

> > ### Author Response · Authors · 2023-08-17
> > **Checking in**
> >
> > Thanks a lot again for your review and feedback. We hope we have addressed your concerns. Please let us know if you have any remaining concerns and questions.

---

> > > ### Comment · Reviewer_uuzi · 2023-08-19
> > > **Thanks for the rebuttal**
> > >
> > > I thank the author(s) for the point-wise response to my comments and questions.
> > >
> > > The responses clarified the points raised in my comments and resolved most of my concerns. I would like to raise my scores to 5, because  up to the current progress, I believe there are still many contents and clarifications (as explained in all rebuttals)  remaining to be rephrased susbstaintially and more concrete evidence should be presented in the paper for elaboration, in order to fully and directly resolve all the raised concerns in the reviews.

---

### Official Review · Reviewer_yCWN · 2023-07-06

**Soundness:** 3 good
**Presentation:** 4 excellent
**Contribution:** 3 good
**Rating:** 6
**Confidence:** 4

**Summary:**

This work studies the design of estimators for the solutions of inverse problems that are adversarially robust, in the sense that they minimize the maximum MSE after being contaminated with an additive and L2 bounded perturbation. The authors show that, for linear estimators and for signals lying on linear subspaces, the optimal solution (i.e. optimally robust) is attained by training the estimator with “Jiterring”, i.e. by minimizing a loss over the randomly perturbed inputs. For more general inverse problems (when the forward operator is not an identity), the authors show that the resulting estimator from Jittering is not optimally robust in general, but their difference can be small. Moreover, they show numerically that this gap is small in practice, thus leading to similar performance (as that obtained by adversarial training) alas with significant increase in computational efficiency.

**Strengths:**

- Neat and elegant idea.
- Very clear presentation - this was a pleasure to read.
- The results are novel and interesting.

**Weaknesses:**

- The results are somewhat limited, and hold for linear estimators with potentially overly simple signal models.

**Questions:**

1. As the authors mention, estimating the robust risk is non-trivial, since this involves the optimization of a non-convex/non-concave problem for non-convex functions $f(\cdot)$ (as in the case of U-net). In light of this, the plots in Fig. 1 are not “exactly” the robust risk, but rather numerical approximations to it (except for the linear case). The authors might wish to clarify this in the caption and/or the description of Fig. 1 in the text.
2. The role of the symmetry of H is not completely clear to me. When describing their results colloquially, the authors mention that, broadly speaking, training linear estimators with Jittering provides an estimator with minimal robust risk if the signal lies on a subspace. However, as written, the formal statement of Theorem 3.1 requires the estimator not only to be linear but also symmetric. So is, is symmetry required for Thm 3.1 to hold, or does symmetry arise as a property of the optimal estimator? I have a related question for Conjecture 3.3 - is symmetry here required, and in particular, $H = UV\Sigma W^T$ need not be symmetric, as written.
3. Their results are stated for $d\to \infty$. It is unclear why this is needed, or how the other dimensions and parameters scale with d.
4. In Eq (3), do the authors mean to require $\lambda \geq max_i \sigma_i^2$?
5. Can the authors comment some more on the choice of norms? More precisely, the authors have focused on the MSE as loss (L2) as well as L2 bounded perturbations. I wonder to what extent their conclusions might extend to other norms. Moreover, it is possible that for different choices, some of the limitations in the analysis (e.g. in proving their conjecture) might be resolved. On the same vein, and because of their signal model, I wonder if there are any connections to the work of [Awasthi, Pranjal, et al. "Adversarially robust low dimensional representations." COLT 2021], that the authors could leverage.

**Limitations:**

Some limitations are commented on throughout the text. Might be nice to stress or clarify them further.

---

> ### Author Rebuttal · Authors · 2023-08-09
>
> We thank the reviewer for the helpful feedback and are very pleased to hear the reviewer enjoyed reading our paper, that the results are novel and interesting, and that the idea is neat and elegant.
> - **Question 1, concerning robust risk vs an approximation of the robust risk**: Thanks for the suggestion, we added "empirical robust risk" to the labels of the plots and included “To approximate the optimal worst-case robust risk, i.e., the minimizer of $R_{\epsilon}$, adversarial training is performed.” in the caption to clarify that we empirically approximate the robust risk, since we can't compute it exactly.
> - **Question 2, the role of symmetry of  $H$**: That is a very good point, it turns out symmetry of $H$ is not required for the main theorem, nor should it be part of the conjecture for general linear inverse problems (section 3.3).
> In the submitted version, we assumed symmetry of $H$. During the rebuttal, we revisited the proof of our main theorem and were able to waive the symmetry assumption completely. This only required minor changes in the proof. Specifically, in the proof of the lower bound in section A.1, we now employ an SVD $H=V \Sigma W^T$, and show that cross-terms between $V$ and $W$ in the robust risk can be lower-bounded suitably such that one can proceed as before.
> - **Question 3, assuming $d \to \infty$**: Our results exactly characterize the optimal estimator for the asymptotic case, and in the asymptotic case there is for example a sharp transition where the estimator maps to zero when the noise energy is equal to the signal energy. For very small d, there is no such sharp transition since the associated random variables do not concentrate well. That said, for moderately large values of d (larger than a constant) our results hold approximately with high probability by utilizing results from high-dimensional probability. We decided to state asymptotic results to provide cleaner expressions. Indeed, our simulations for the linear model show that the analytical formula is already accurate for moderate dimensions (d > 25).
> _Regarding the other variables_: In the limit subspace dimension $d \to \infty$, the embedding space $n > d$ is similarly treated in this limit. The energies $\sigma_c^2$ and $\sigma_z^2$, however, do no not depend on the dimensions.
> - **Question 4, question on Eq (3)**: $\lambda \geq \sigma_i^2$ is used as an abbreviation of $\lambda \geq \max_i \sigma_i^2$. We included the maximum in the revision for clarification.
> - **Question 5, considering other norms:** We consider the $\ell_2$-norm for the loss function and perturbations, since they are most relevant for inverse problems ($\ell_2$-norm measures signal energy).
> We would certainly be interested in extending our theory to other norms. However our results do not generalize in a straightforward manner to other norms. Specifically, it is unclear how to generalize Lemma 1 of the appendix, which reformulates the robust risk optimization problem to a tractable one with respect to a single variable.
> However, while it's unclear how to obtain a precise characterization, some more general predictions can be made. As an example, the transition predicted to the zero-estimator, predicted by Theorem 1, also exists $\ell_p$-type perturbations ($p \geq 2$). From our understanding, the work on robust PCA the reviewer pointed us to studies efficient approximation algorithms for finding $\ell_p$-adversarially robust subspaces (but not actually characterizing the optimal ones). Moreover, finding these subspaces consists of finding optimal projection matrices onto subspaces, whereas we consider general reconstruction matrices for solving linear inverse problems. The paper is very interesting and we’ll investigate further connections. We appreciate the reference and add a discussion of this work to the related work section of our paper.
>
> We really appreciate your feedback, which has improved the paper, in particular it led us to generalize our results to non-symmetric H. We hope this addresses your concerns. If so, we would appreciate it if you can consider increasing your score. Of course we are happy to address any further questions or comments you may have.

---

> > ### Comment · Reviewer_yCWN · 2023-08-13
> > **Thank you for your responses**
> >
> > I thank the authors for carefully considering and addressing my comments and questions. I've decided to increase my score to 6 - my main limitation to increase the score further is the limitation of their results which address only very simple cases. If accepted, I encourage the authors to make your assumptions clear (remove the unnecessary assumption on symmetry, state clearly that the results hold asymptotically, and provide the extended results on approximations with high probability if possible).

---

### Official Review · Reviewer_koTN · 2023-07-07

**Soundness:** 3 good
**Presentation:** 3 good
**Contribution:** 3 good
**Rating:** 7
**Confidence:** 3

**Summary:**

The authors study the effectiveness of jittering in the setting of inverse problems, specifically considering denoising, compressive sensing, and deconvolution problems. Jittering is a well-known regularization technique for classification problems. The authors prove in the linear setting that the robust risk estimator can be probed to be the optimal estimator when learned with jittering for Gaussian subspace denoising. In addition, the authors demonstrate experimentally jittering is effective at improving the robustness of compressive sensing and image deconvolution, even though it yields suboptimal estimates.

**Strengths:**

1). The theoretical contribution is straightforward but has a strong contribution in the denoising setting. Even though the authors studied a linear model they were able to provide insights in the non-linear model (U-net) that was studied empirically. Refer to Figure 3 (Middle Plot), Cor.3.2 can accurately predict the optimal jittering level.

2). The authors extend theoretical contribution for general inverse problems, section 3.3.

3). The authors do a thorough investigation on the effectiveness of jittering empirically and theoretically for denoising inverse problems.

**Weaknesses:**

1). The motivation of the paper seems a bit unclear, it appears to be motivated by results from classification problems rather than image reconstruction results in inverse problems. This approach seems slightly ad-hoc because the authors choose a popular regularization technique for classification problems and tested if it was effective for inverse problems without stating their intuition for it being effective or being of interest to the community.

2). Section 4.2 lacks some metrics, specifically, Section 4.2 Figure 4 states "Jittering yields robust estimators, but at the same time yields smoother reconstructions.". This claim seems pretty strong considering it is not reported over a large test set with appropriate metrics for image reconstruction.

3).  The section for 3.3 appears disconnected from section 4, it's unclear whether the linear model studied in this section (3.3) is representative of the behavior of the non-linear model utilized in section 4.





Minor Weakness-
Typo Figure 3- "The jittering estimators estimators are similarly robust as adversarial
training (Figure 1), but attain lower standard risks (right panel)" Repeated word estimators estimators

**Questions:**

1). Could you provide metrics for a test set confirming jittering yields smoother reconstructions?

2). Could the authors justify why they studied a linear model instead a non-linear model, a model more similar to a U-net?

3). Could the authors explain the connection of section 3.3 to experiments in Section 4? Does the linear model exhibit enough similar behavior as the non-linear model to justify it as an appropriate model for general linear inverse problems? If so, could you please provide some numerical evidence?



**Limitations:**

I do not see any negative societal impacts of this work.

---

> ### Author Rebuttal · Authors · 2023-08-09
>
> We thank the reviewer for the helpful feedback and for noting that we make a strong contribution for denoising, and that even though 'they study a linear model they were able to provide insights in the non-linear model (U-net) that was studied empirically'.
>
> Comments on the weaknesses and answers to the questions:
>
> **Weakness (1), clarifying the motivation of the paper.** The main motivation for the paper is to understand whether neural networks for signal/image reconstruction can be trained efficiently to be worst-case robust. To address this question, we first characterize the worst-case optimal estimator for a linear setup. This is the main technical contribution. While the worst-case optimal estimator is intuitive, showing optimality is non-trivial.
>
> We then investigate whether jittering, a simple regularization technique that adds isotropic Gaussian noise during training, is effective for learning worst-case robust estimators for inverse problems, motivated by an ongoing discussion in the community on whether jittering is effective or not specifically for signal/image reconstruction problems. Specifically, Genzel et al. (2022) finds that jittering is effective for MRI and CT, whereas Gandikota et al. (2022) report suboptimality for deconvolution.
>
> Our work shows that jittering is provably effective for inverse problems, which we consider to be of great interest for the community given that it is computationally so much cheaper than adversarial training.
>
> We'll clarify this motivation in the paper.
>
> **Weakness (2) and Question (1), on metrics capturing the smoothing effect:** During the rebuttal, we measured the smoothness using the total variation (TV) norm of reconstructed images for the deconvolution problem (same networks as in Figure 1; 2k test images). The results are in the attached pdf and confirm our observation that jittering yields to smoother reconstructions: The TV-norm of reconstructions using networks trained via jittering is generally smaller than those of standard or adversarial training.
>
> **Question (2), why doing theory for a linear model and not for a U-net**: We certainly would like to extend our theory to deep neural networks in particular a U-net or the like, but right now the theoretical tools for characterizing optimal network estimators of the complexity of U-nets do not exist to the best of our knowledge. The vast majority of results for neural networks is actually for linear networks, and for networks in the neural-tangent-regime that behave like an associated linear model. Our setup is already very difficult to treat analytically, since we have a minimization followed by an expectation followed by a maximization. That said, as we demonstrate empirically the insights carry over to neural networks and serve as an important first step towards analyzing nonlinear networks. Indeed, many theoretical results that are emerging (NTK, power method analysis in first few iterations) rely on understanding of appropriate linear networks for their nonlinear analysis. We hope to build upon this first step in future work.
>
> **Weakness (3) and Question (3), on the connection of 3.3 (theory for a linear model) to section 4 (experimental results):**
> While our theory is for a linear model, our experiments are for real-world imaging problems. We find a strong agreement of the theoretical results for the linear model and the real-world simulations for the U-net:
> - Jittering empirically yields optimal worst-case robust U-Net denoisers, as proven for the linear estimator (Fig. 1).
> - Jittering can be suboptimal for inverse problems beyond denoising, which is explained by our theory for general linear inverse problems in section 3.3 (Fig. 2 and Fig. 1).
> - The optimal jittering noise levels for U-net-denoisers are accurately predicted by theory (Fig. 3).
> - Adversarial training of U-nets even reproduces the theoretically predicted extreme behavior for large perturbations (obtaining U-nets mapping everything to zero, Fig. 8).
>
> Please let us know if that addressed your concerns, if the clarifications change your final score, and if you have any further questions or comments.

---

> > ### Comment · Reviewer_koTN · 2023-08-14
> > **Thank you for the response**
> >
> > I thank the authors for carefully considering and addressing my questions/concerns. After reading the rebuttal, I would like to increase my score to a 7. I believe the paper is a strong contribution to the community of researchers interested in inverse problems.

---

### Author Rebuttal · Authors · 2023-08-09

We thank the reviewers for their helpful feedback. In this post, we address one of reviewer *koTN*'s questions.

**Metrics quantifying the smoothing effect**: During the rebuttal, we measured the total variation (TV) norm of reconstructed images for the deconvolution task over a large test dataset. The results are presented in the attached pdf and confirm our observation that jittering yields smoother reconstructions compared to adversarial and standard training.

---

### Decision · Program_Chairs · 2023-09-21

**Decision:**

Accept (poster)

**Comment:**

This paper studies adversarially robust estimators to inverse problems by Jittering ie minimizing the loss after the input has been perturbed.
The idea is novel, the method is reasonable and seems to work well. The reviewers liked the paper and asked reasonable questions that were addressed in the rebuttal. So this is a clear accept.